



# Spatial patterns of biphasic ectoenzymatic kinetics related to biogeochemical properties in the Mediterranean Sea.

France Van Wambeke[1], Elvira Pulido[1], Julie Dinasquet[2,3], Kahina Djaoudi[1,4], Anja Engel[5], Marc Garel[1], Sophie Guasco[1], Sandra Nunige[1], Vincent Taillandier[6], Birthe Zäncker[6,7], Christian Tamburini[1].

[1]Aix-Marseille Université, CNRS/INSU, Université de Toulon, IRD, Mediterranean Institute of Oceanography (MIO) UM 110, 13288, Marseille, France
[2]Marine Biology Research Division, Scripps Institution of Oceanography, UCSD, La Jolla, USA
[3]Sorbonne Universités, UPMC University Paris 6, Laboratoire d'Océanographie Microbienne (LOMIC), Observatoire Océanologique, 66650, Banyuls/mer
[4]Molecular and Cellular Biology, The University of Arizona, Tucson, USA
[5]GEOMAR – Helmholtz-Centre for Ocean Research, Kiel, Germany
[6]CNRS, Sorbonne Universités, Laboratoire d'Océanographie de Villefranche (LOV), UMR7093, 06230 Villefranche-sur-Mer, France
[7]The Marine Biological Association of the UK, Plymouth, United Kingdom

*Correspondence to*: F. Van Wambeke (france.van-wambeke@mio.osupytheas.fr)

**Abstract.** Prokaryotic ectoenzymatic activity, abundance and heterotrophic production were determined in the Mediterranean Sea, within the epipelagic and the upper part of the mesopelagic layers. The Michaelis-Menten kinetics were assessed, using a range of low (0.025 to 1 µM) and high (0.025 to 50 µM) concentrations of fluorogenic substrates. Thus, Km and Vm parameters were determined for both low and high affinity systems for alkaline phosphatase (AP), aminopeptidase (LAP) and β−glucosidase (βGLU). Based on the constant derived from the high AP affinity system, *in-situ* hydrolysis rates of N-protein contributed of 48% ± 30% for the heterotrophic prokaryotic nitrogen demand within epipelagic waters and of 180% ± 154% within deeper layers. LAP hydrolysis rate was higher than bacterial N demand only within the deeper layer, and only based on the high affinity system. Although ectoenzymatic hydrolysis contribution to heterotrophic prokaryotic need was high in terms of N, but low in terms of C. Based on a 10% bacterial growth efficiency, the cumulative hydrolysis rates of C-proteins and C-polysaccharides contributed to a small part of the heterotrophic prokaryotic carbon demand, on average 2.5% ± 1.3% in the epipelagic layers. This study notably points out the biases in current and past interpretation of the relative activities differences among the 3 tested enzymes, in regard to the choice of added concentrations of fluorogenic substrates. In particular, enzymatic ratios LAP/βGLU, as well as some trends with depth, were different considering activities resulting from the high or the low affinity system.

## 1. Introduction

Most of the organic matter being in the state of high molecular weight material, its hydrolysis by ectoenzymes plays an important role in the degradation, utilization and mineralization processes in aquatic environments, but also in nutrient's regeneration (Hoppe, 1983; Chróst, 1991). Whether or not the ectoenzymatic activity must be considered as a limiting rate in organic matter remineralization is subject of debate as hydrolysis and consumption of hydrolysis products are not always coupled (e.g. for instance Smith et al., 1992). Bacterial ectoenzymatic hydrolysis is usually determined using fluorogenic substrates (Hoppe, 1983) which, upon cleavage by ectoenzymes, trigger the release of a fluorescent by-product. The increase of the latter is monitored by fluorimetry over time, allowing quantification of ectoenzymatic hydrolysis rates. Kinetic experiments are time-


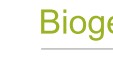
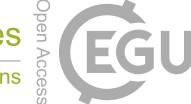

consuming and most studies reporting ectoenzymatic activity seldomly examine enzyme kinetic
patterns, but only assess activity based on one substrate concentration assumed to be saturating.
Baltar et al. (2009b) report that among 17 published studies, 12 used a single range of substrate

concentrations, varying from 0.02 to 1000 μM (with a median of 50 μM), and only 5 considered
variable ranges. Within these 5 studies, for the concentration kinetic, the minimum concentration
used was 50 nM at the lowest, and was more generally between 1 and 5 μM, and the highest
concentration used ranged between 5 μM and 1200 μM, median 200 μM. In the Mediterranean Sea,
a compilation of data by Zaccone and Caruso (2019) shows that among 22 studies, 6 used a single

concentration (that they assumed to be saturating) with a median of 125 μM for MCA-leu and 50
μM for MUF-P. While the other studies assessed kinetics systematically, the range of
concentrations was again highly variable (min 0.025 - 200 median 0.1 μM, max 1 - 4000 median 20
μM). However, the combination of: i) non-specific target of the enzymes, ii) the heterogeneity of
enzymatic systems among a single species, iii) the diversity of present species and iv) the

concentration of surrounding substrates, will result in multiphasic kinetics (Chróst, 1991; Arnosti,
2011; Sinsabaugh and Shah, 2012 and ref therein). It is well known that ectoenzymes can be
produced by a diversity of microorganisms. Their activity depends on a patchy distribution of
natural substrates, and a variety of natural (potentially unknown) molecules hydrolyzed by the same
enzymes that hydrolyze the added fluorochrome, all probably having different affinities for the

enzyme. Indeed, cell-specific activities and types of activities were shown to be extremely variable
among 44 isolated strains, cultivated in batch cultures and sampled during exponential phase
(Martinez et al., 1996). Arrieta and Herndl (2001) assessed the diversity of marine bacterial β-
glucosidases from a natural community separated using capillary electrophoresis zymography and
showed that they had different Km and Vm. Biphasic kinetic systems have been described in areas

where increasing gradients of polymeric material are expected due to the high concentration of
particles; e.g. near bottom water and sediment for aminopeptidase (Tholosan et al., 1999), in a
shallow bay for phosphatases (Bogé et al., 2013). Moreover, in the water column different kinetic
systems were also observed and are generally attributed to attached or free-living bacteria having
different affinities for substrates, k-strategists-oligotrophic bacteria (with low Km and low Vm) or

r-strategists/copiotroph bacteria (with high Km and Vm, Koch, 2001). At depth, the simultaneous
presence of more refractory DOM with recent and freshly sedimenting particles, suggests also
possible multiphase kinetics for ectoenzymatic activity. Varying kinetic parameters were also
attributed to 'free' (i.e. related to free-living bacteria) *versus* 'attached" heterotrophic bacteria, the
latters highlighting higher both Vmax and Km, i.e. adapted to cope with high substrate

concentrations (Unanue et al., 1998). Due to the soluble nature of the fluorochrome released, these
estimates are based on size fractionation prior the incubations, which biases ectoenzymatic
activities in each size fraction, due to filtration artifacts and disruption of trophic relationships
between primary producers, heterotrophic bacteria, regenerative protozoans and osmotrophs.
Nevertheless, most studies have shown that cell-specific ecotoenzymatic activities on aggregates are

~10 fold higher than those of the surrounding assemblages (decaying bloom, Martinez et al., 1996).
Furthermore, carbon budgets have shown that the prokaryotes attached to aggregates are likely a
source of by-products for free-living prokaryotes, thanks to their ectoenzymatic hydrolysis (Smith
et al., 1992). The use of bulk water concentration kinetics allows the determination of different
enzymatic kinetics without disturbing relationships between free/attached prokaryotes and

DOM/POM interactions during the incubations.

In the Mediterranean Sea, elemental C/N/P ratios of nutrients and organic matter are the object of
particular attention and debates, especially when it comes to the origin of P-deficiency and DOC



accumulation (Thingstad and Rassoulzadegan, 1995; Krom et al., 2004). The export of organic carbon in dissolved *vs.* particular forms is related to the P-limitation status within surface layers

(Guyennon et al., 2015). Therefore, models used to compute C, N and P budgets in the Mediterranean Sea must consider non-Redfield C/N/P stoichiometry (Baklouti et al., 2006). The interaction between different enzymes has been largely studied in the Mediterranean Sea (Zaccone and Caruso, 2019) due to the particular role of this elemental stoichiometry. Indeed, the epipelagic layers are P or N-P limited during most periods of water mass stratification, and ectoenzymes

providing P and N sources from organic matter, such as aminopeptidase and phosphatase have been intensively studied as indicators of these limitations (Sala et al., 2001; Van Wambeke et al., 2002). However, the potential bias introduced by multiple kinetics, when comparing different types of ectoenzymes, is still poorly understood.

In this study, we investigated in the Mediterranean Sea, the kinetics of three series of enzymes targeting

proteins, phospho-mono esters and carbohydrates (aminopeptidase, alkaline phosphatase and β-D – glucosidase, respectively) in relation to the elemental stoichiometry of particulate and dissolved organic matter. We have paid particular attention to the use of a wide range of substrates concentrations to evaluate potential multiphasic kinetics. Our aim was to study the effects of the respective activities of the ectoenzymes in relation to the quality of the available organic matter, below the productive layer

and above the deep Mediterranean waters. We were especially interested in the Levantine Intermediate Waters (LIW), a typical Mediterranean water mass that spreads from ~200 to 700 m, characterized by a local maximum of salinity and a local minimum of dissolved oxygen concentration (e.g. Kress et al., 2003; Malanotte-Rizzoli et al., 2003). This study focuses on the open waters of the western Mediterranean Sea and Ionian Sea, considering four water layers: surface (generally P or N limited

during the stratification period), the deep chlorophyll maximum layer (coinciding with nutricline depths), the LIW and below the LIW. Finally, we discuss the biases in interpretation of past and current enzymatic kinetic, potentially induced by the reduced range of used substrate concentrations. The data used in this ms are also developed in another article (Van Wambeke et al, in prep) of the special issue in which all biogeochemical fluxes (phytoplankton and heterotrophic bacterial P and N demand, N2

fixation rates, aminopeptidase, phosphatase activities) within the mixed layers will be compared with wet and dry N and P atmospheric fluxes thanks to an exceptional simultaneous measurements of all these fluxes during the same cruise.

## 2. Materials and Methods

### 2.1 Sampling strategy

The PEACETIME cruise (doi.org/10.17600/15000900) was conducted in the Mediterranean Sea, from May to June 2017, along a transect extending from the Western Basin to the center of the Ionian Sea (25°S 115 E – 15°S, 149°W, Fig. 1). For details on the cruise strategy, see Guieu et al. (2020). Stations of short duration (< 8 h, 15 stations named SD1 to SD10, Fig. 1) and long duration (5 days, 3 stations named TYR, ION and FAST) were sampled. Generally, at least 3 casts were

conducted at each short station. One focused on the epipelagic layer (0 - 250 m), and the second one focused on the whole water column. Both were sampled with a standard, classical-CTD rosette equipped with a sampling system of 24 Niskin bottles (12 L), and a Sea-Bird SBE9 underwater unit equipped with pressure, temperature (SBE3), conductivity (SBE4), fluorescence (Chelsea Acquatracka) and oxygen (SBE43) sensors. The third cast, from surface to bottom was performed

under 'trace metal clean conditions' using a second instrumental package (called TMC-rosette) mounted on a Kevlar cable and equipped with Go-Flo bottles that were sampled in a dedicated trace





metal free-container. The long stations were abbreviated as TYR (situated in the center of the Tyrrhenian Basin), ION (in the center of the Ionian Basin), FAST (in the western Algerian Basin). Long stations were selected using satellite imagery, altimetry and Lagrangian diagnostics and
expected events of rain (Guieu et al., 2020). At these stations, repeated casts were performed during at least 5 days, alternating CTD- and TMC- rosettes.

The water sampled with the conventional CTD-rosette, was used for measuring heterotrophic prokaryotic production (BP), heterotrophic prokaryotic abundances (BA), ectoenzymatic activity (EEA), chlorophyll stocks, particulate organic carbon (POC), nitrogen (PON) and phosphorus
(POP) and dissolved organic carbon (DOC). The TMC-rosette was used for dissolved inorganic nitrogen (DIN) and phosphorus (DIP), dissolved organic nitrogen (DON) and phosphorus (DOP).

Besides measurements of the biogeochemical variables, BP and ectoenzymatic activities described below, other data presented in this paper include hydrographic properties (T, S, $O_2$), total chlorophyll a (Tchl$a$) concentrations for which detailed protocols of analysis and considerations for
methodology are available in Taillandier et al. (2020), Guieu et al. (2020), and Maranon et al. (2020)

We focused on 4 layers of the water column: two were in the epipelagic waters: surface (5 m, named 'surf'), and the deep chlorophyll maximum layer, localized by the *in vivo* fluorescence continuously measured by the instrumental package during downcasts (named 'dcm'); and two
deeper layers: one in the LIW (localized by maximum salinity and minimum oxygen properties during downcasts, named 'liw'), and second sampled at 1000 m (the limit between meso and bathypelagic waters), except at 2 stations (FAST, 2500 m; ION, 3000 m) named 'mdw'. Table 1 resumes the sampled stations with their localization, characteristics and depths of 'dcm', 'liw' and 'mdw' layers sampled.

**2.2 Biochemistry**

Nitrate (NO3), nitrite (NO2), and orthophosphate (DIP) concentrations were determined on a segmented flow auto-analyzer (AAIII HR Seal Analytical) according to Aminot and Kérouel (2007). The quantification limits were 0.05 µM for NO3, 0.01 µM for NO2 and 0.02 µM for DIP. The dissolved organic pools, DON and DOP, were determined after high-temperature (120 °C)
persulfate wet oxidation mineralization (Raimbault et al., 1999). Twenty ml of water sample were filtered on a 0.2 µm PES membrane and collected into 25 ml glass flasks. Samples were immediately poisoned with 100 µl $H_2SO_4$ 5N and stored in the dark until analysis in the laboratory. For DON and DOP, filtered samples were then collected in Teflon vials adjusted at 20 ml for wet oxidation. Nitrate and phosphate formed after the wet persulfate oxidation, corresponding to total
dissolved pool (TDN and TDP), were then determined as previously described for the dissolved inorganic pools. DON and DOP were obtained by the difference between TDN and DIN, and TDP and DIP, respectively. The limits of quantification were 0.5 and 0.02 µM for DON and DOP, respectively.

The particulate pools (PON, POP) were determined using the same wet oxidation method
(Raimbault et al., 1999). A volume of 1.2 L was collected from Niskin bottles in polycarbonate bottles and directly filtered through a pre-combusted (450 °C, 4 h) glass fiber filter (Whatman 47mm GF/F) and kept frozen at -20°C until return to the laboratory. For the analysis, filters were placed in Teflon vials with 20 mL of ultrapure water (Milli-Q grade) and 2.5 mL of the wet oxidation reagent for mineralization. Nitrate and orthophosphate produced were analyzed as





described previously. The limits of quantification were 0.02 and 0.001 µM for PON and POP, respectively.

Within epipelagic, nutrient depleted layers DIP and NO3 were also determined using the LWCC method (Zhang and Chi, 2002) in which the sensitivity of the spectrophotometric measurement is improved by increasing the optical path length of the measurement cell to 2.5 m. For DIP,

quantification limits were decreased down to 0.8 nM and the response is linear up to about 150 nM, for NO3, quantification limits decreased to 9 nM.

Samples for dissolved organic carbon (DOC) were filtered through two pre-combusted (24 h, 450°C) glass fiber filters (Whatman GF/F, 25 mm) using a custom-made all-glass/Teflon filtration syringe system. Samples were collected into pre-combusted glass ampoules and acidified to pH 2

with phosphoric acid ($H_3PO_4$). Ampoules were immediately sealed until analyses by high temperature catalytic oxidation (HTCO) on a Shimadzu TOC-L analyzer (Cauwet, 1999). Typical analytical precision is ± 0.1-0.5 (SD) or 0.2-0.5% (CV). Consensus reference materials (http://www.rsmas.miami.edu/groups/biogeochem/CRM.html) was injected every 12 to 17 samples to insure stable operating conditions. Particulate organic carbon (POC) was measured using a CHN

analyzer and the improved analysis proposed by Sharp (1974).

Samples (20 ml) for total hydrolysable carbohydrates (TCHO) > 1 kDa were filled into precombusted glass vials (8 h, 500°C) and stored at -20°C until analysis. Samples were desalinated with membrane dialysis (1 kDa MWCO, Spectra Por) at 1°C for 5 h. Samples were hydrolyzed for 20 h at 100°C with 0.8 M HCl final concentration with subsequent neutralization using acid

evaporation ($N_2$, for 5 h at 50°C). TCHO were analysed using high performance anion exchange chromatography with pulsed amperometric detection (HPAEC-PAD) which was applied on a Dionex ICS 3000 ion chromatography system (Engel and Händel, 2011). Two replicates per TCHO sample were analyzed.

Total hydrolysable amino acids (TAA) were determined from 5 mL of sample which was filled into

precombusted glass vials (8 h, 500°C) and stored at -20°C. Samples were measured in duplicates. The samples were hydrolyzed at 100°C for 20 h with 1 mL 30% HCl (Suprapur®, Merck) to 1 ml of sample added and neutralized by acid evaporation under vacuum at 60°C in a microwave. Remaining acid was removed with water. Samples were analyzed by high performance liquid chromatography (HPLC) using an Agilent 1260 HPLC system following a modified version of

established methods (Lindroth and Mopper, 1979; Dittmar et al., 2009). Prior to the separation of 13 amino acids with a $C^{18}$ column (Phenomenex Kinetex, 2.6 µm, 150 x 4.6 mm), in-line derivatization with o-phthaldialdehyde and mercaptoethanol was performed. A gradient with solvent A containing 5% acetonitrile (LiChrosolv, Merck, HPLC gradient grade) in sodium dihydrogen phosphate (Suprapur®, Merck) buffer (pH 7.0) and solvent B being acetonitrile was used for analysis. A

gradient from 100% solvent A to 78% solvent A was produced in 50 min.

### 2.3 Bacterial production

Bacterial production (BP, *sensus stricto* referring to prokaryotic heterotrophic production) was determined onboard using the microcentrifuge method with the $^3$H- leucine ($^3$H-Leu) incorporation technique (Smith and Azam, 1992) within epipelagic waters, and the filtration technique for deep

waters, as the centrifuge technique is limited to incubation volumes of 1.5 mL and was not sufficiently sensitive in deep waters. In epipelagic waters, triplicate of 1.5 mL samples and a killed control with trichloracetic acid (TCA, 5% final concentration) were incubated with a mixture of



[4,5-³H]-leucine (Amersham, specific activity 112 Ci mmol⁻¹) and non-radioactive leucine at final
concentrations of 7 and 13 nM, respectively. Samples were incubated in the dark at the respective *in
situ* temperatures for 1- 4 h. On 9 occasions, we checked that the incorporation of leucine was linear
with time. Incubations were ended by the addition of TCA to a final concentration of 5%, followed
by three runs of centrifugation at 16000 g for 10 minutes. Bovine serum albumin (BSA, Sigma, 100
mg L⁻¹ final concentration) was added before the first centrifugation. After discarding the
supernatant, 1.5 mL of 5% TCA was added before the second centrifugation, and for the last run,
after discarding the supernatant, 1.5 mL of 80% ethanol was added. The ethanol supernatant was
discarded and 1.5 mL of liquid scintillation cocktail (Packard Ultimagold MV) was added. For 'liw'
and 'mdw' layers, 40 mL samples were incubated in the dark for up to 12 hours at the *in situ*
temperature (triplicate live samples and one formalin-fixed control), fixed with formalin (2% final
concentration, using 10 nM [4,5-³H]-leucine. After filtration of the sample through 0.2 μm
polycarbonate filters, 5% TCA were added for 10 minutes, and then the filter was rinsed with an
additional 10 mL rinse with 5% TCA and a final 80% ethanol rinse.

For both types of samples (centrifuge tubes and filters) the radioactivity incorporated into
macromolecules was counted in a Packard LS 1600 Liquid Scintillation Counter on board the ship.
A factor of 1.5 kg C mol leucine⁻¹ was used to convert the incorporation of leucine to carbon
equivalents, assuming no isotopic dilution (Kirchman, 1993), as checked from occasional
concentration kinetics. Standard deviations associated with the variability between triplicate
measurements averaged 8% and 25% for BP values estimated with the centrifugation (higher
activities surface layers) or the filtration technique (lower activities, deep layers), respectively.

### 2.4 Ectoenzymatic activities

Ectoenzymatic activities were measured fluorometrically, using fluorogenic model substrates that
were L-leucine-7-amido-4-methyl-coumarin (Leu-MCA), 4 methylumbelliferyl – phosphate (MUF-
P), 4 methylumbelliferyl – βD-glucopyranoside (MUF-βglu) to track aminopeptidase activity
(LAP), alkaline phosphatase activity (AP), and β−glucosidase activity (βGLU), respectively
(Hoppe, 1983). Stocks solutions (5 mM) were prepared in methycellosolve and stored at −20°C.

Release of the products of LAP, AP and βGLU activities, MCA and MUF, were followed by
measuring increase of fluorescence (exc/em 380/440 nm for MCA and 365/450 nm for MUF,
wavelength width 5 nm) in a VARIOSCAN LUX microplate reader. The instrument was calibrated
with standards of MCA and MUF solutions diluted in < 0.2 μm filtered and boiled seawater. For
measurements, 2 mL of unfiltered seawater samples were supplemented with 100 μL of a
fluorogenic substrate solution diluted so that different concentrations were dispatched in a black 24-
well polystyrene plate in duplicate (0.025 to 50 μM). Three plates were filled per layer analyzed
with the different substrates MUF-P, MCA-leu and MUF-βglu. Incubations were carried out in the
dark in thermostatically controlled incubators reproducing *in situ* temperature ranges. Incubations
lasted up to 24 h long with a reading of fluorescence every 1 to 3 h, depending on the intended
activities. The rate was calculated from the linear part of the fluorescence *versus* time relationship.
Boiled-water blanks were run to check for abiotic activity. From varying velocities obtained, we
determined the parameters Vmax (maximum hydrolysis velocity) and Km (Michaelis-Menten
constant which reflects enzyme affinity for the substrate) by fitting the data using a non-linear
regression on the following equation: $V = Vmax \times S/(Km + S)$, where V is the hydrolysis rate and S
the fluorogenic substrate concentration added. We determined Vm and Km using 2 series of
substrate concentrations: Vm50 and Km50 were calculated using a range of 11 concentrations



(0.025, 0.05, 0.1, 0.25, 0.5, 1, 2.5, 5, 10, 25 and 50 µM) in duplicate and Vm1 and Km1 were calculated using a restricted range of substrate concentrations up to 1 µM (0.025, 0.05, 0.1, 0.25, 0.5, 1 µM) in duplicate. We used the term 'ectoenzyme' and considered that it included all types of
enzymes found outside the cell, including enzymes fixed on cells (on external membranes, or within the periplasmic space), and free-dissolved enzymes, to broadly encompass all enzymes located outside intact cells regardless of the process by which such enzymes entered the environment.

We used an approach similar to Hoppe et al. (1993) to compute *in situ* hydrolysis rates for LAP and βGLU (we did not make this calculation for AP as this is developed in a companion paper from this
special issue in Pulido-Villena et al., in prep). We used total carbohydrates (TCHO) and total aminoacids (TAA) data and assuming they could be representative of dissolved combined carbohydrates and dissolved proteins, respectively. *In situ* hydrolysis rates of LAP and βGLU were determined using molar concentrations of TAA and TCHO used as substrate concentration in the Michaelis-Menten kinetics, respectively. These rates were calculated based on Vm1 Km1, in one
hand and on Vm50 and Km50, on the other hand. The transformation of *in situ* hydrolysis rates expressed in nmol L$^{-1}$ h$^{-1}$ were then transformed in carbon units using C per mole TCHO, C per mole TAA, and in nitrogen units using N per mole TAA, as the molar distributions of TAA and TCHO were available.

### 2.5 Statistics

Trends with depth were estimated using a depth variation factor (DVF) calculated as the mean of pooled 'surf'and 'dcm' data divided by the mean of pooled 'liw' and 'mdw' data. This decrease (or increase), was considered as significant after a t-test comparing both series of data. The type of t test used depended on the result of a preliminary F-test checking for variance. The prism 4 (Graph Pad software, San Diego, USA) was used to perform nonlinear regressions on Michaelis-Menten
kinetics. Means are cited ± standard errors. Correlations between variables were examined after log transformation of the data.

### 3.Results

### 3.1 Hydrological situations.

The sampled stations have basins and latitude characteristics that were superimposed on a changing
the seasonal pattern. Lower surface temperatures (14 - 17°C) were thus measured at ST1 to ST3, sampled in the beginning of the cruise at higher latitudes, and higher temperatures at ST10 (21.6°C) were sampled at the end of the cruise. T/S diagrams presented profiles characteristic of the different basins and water masses (Fig. 2). Modified Atlantic Waters (MAW) are characterized by low salinity below the seasonal thermocline; this property is stretched in the westernmost stations, then
progressively relaxes on eastern station, revealing an eastward circulation in the Algerian Basin and a dispersion in the connected basins (northwestern Mediterranean, Tyrrhenian, and Ionian Seas). Levantine intermediate waters (LIW) lying at depths of 200 to700 m are characterized by local salinity maximum in the T/S diagram. This property is pronounced in the eastern stations and progressively lowered on the western stations, revealing an opposite circulation pattern to the
MAW. The western and eastern Mediterranean Deep waters (WMDW and EMDW, respectively) are formed respectively in the northwestern Mediterranean and in the Adriatic-Aegean Seas; they occupy depth ranges below the LIW and are clearly separated by the bathymetric shallow sill at the Sicily strait. The WMDW T/S characteristics (potential temperature 12.91°C, salinity 38.48) were less salty and colder than in the EMDW water mass observed in the Ionian Sea (potential





temperature13.43°C, salinity 38.73). The core of LIW is characterized by lower oxygen content than its surrounding water masses, shallower (MAW) and deeper (WMDW and EMDW). The core of LIW becomes colder, less salty and deepened along its main trajectory toward the West. We thus presented all the figures/tables in the order ST10, FAST, ST1, ST2, ST3, ST4, ST5, TYR, ST6 and ION, according to the expected circulation of the LIW (from the right to the left). The 'liw' samples

were situated in a range of 100 m above - 150 m below the core of LIW but were still in its water mass at all stations. The deeper layer sampled 'mdw' corresponded to the top of WMDW water mass in the Western Basin, was inside the modified WMDW in the Tyrrhenian Basin, and was inside the EMDW in the Ionian Basin. Note however that ST10 is within an anticyclonic eddy, and ST3 is influenced by the water dynamics along the continental slope of Sardinia.

**3.2 Biogeochemical situation**

Nitrate and phosphate were depleted in the surface layers, exhibiting concentrations below the detection limits of the classical methods (0.01 µM, table S1). Using the LWCC technique, however, DIP was detectable (Table S1) and ranged between 4 to 17 nM at 5 m depth. The depth of nitracline (roughly estimated from the depth where $NO_3$ reaches 50 nM) ranged from 30 to 85 m (Table 1).

Phosphaclines were deeper in the Eastern basin, with greater differences between the depths of phosphacline and nitracline, particularly at ST 6 and ION. Chlorophyll content ranged from 18.7 to 35 mg Tchl$a$ m$^{-2}$ at ST 6 and ST1, respectively (Table 1). The depth of the deep chlorophyll maximum ranged from 49 to 83 m in the Western basin, with no particular trend in the Tyrrhenian Sea and its deepest location was in the Ionian Sea (105 m at ION).

The highest DOC values were generally observed within surface layers, and decreased by approximatively 10 µM between layers, the depth variation factor (DVF) ranged from x1.2 to x1.6. For DON, the DVF was close to that of DOC, ranging from x1.2 to x1.8. Including the 4 layers, means of DOC/DON and DOC/DOP molar ratios were 14 ± 2 and 2112 ± 1644, respectively with no significant trend with depth due to the variability within stations. Means of TAA were stable

between 'surf' and 'dcm' layers, around 210 nM, as the maximum values were either observed within the surface, or within the 'dcm' layers (Table S1, Fig. S1a). At all stations TAA rapidly decreased between the epipelagic and deeper waters ($p < 0.001$). The mean DVF of TAA (x3.4) was twice higher than that of DON (x1.5) and this trend was confirmed by contribution of TAA-N to DON which ranged from between 3 and 9% within the 'surf' and the 'dcm' layers to between 1.6

and 4.6% at the 'liw' and 'mdw' layers (Fig. S1a) and this contribution decreased significantly with depth ($p < 0.001$). TCHO ranged from 111 to 950 nM. The contribution of TCHO-C to DOC ranged from 1.3 to 9.7% according stations and depths. If epipelagic TCHO values were relatively constant among the different stations (means 595 ± 43 nM in 'surf', 351 ± 73 nM in 'dcm', Table S1) the deeper TCHO distributions varied between station as noted for 'mdw' layer in ST5, TYR and ST6

in the Tyrrhenian Sea where the highest values were obtained (Fig. S1b). At some other stations, 'mdw' layers did not display the highest TCHO values but increased compared to' liw' layers. So that, at 6 stations out of 10, a minimum TCHO value was obtained within 'liw' layer (Fig. S1b). TCHO-C to TAA-C ratio were 4.3 ± 1, 2.5 ± 0.3, 5.5 ± 3.5 and 15 ± 14 within 'surf', 'dcm', 'liw' and 'mdw' layers, respectively, increasing significantly with depth ($p < 0.02$) and exhibiting

particularly high ratios within the Tyrrhenian sea 'mdw' layer (ST5 48 TYR 24 , ST6 27).

**3.3 Ectoenzymatic activities – kinetic trends**

The ectoenzymatic activities were determined using large trophic conditions and over a wide range of substrate concentrations ranging from 0.025 to 50 µM. Different types of kinetics were obtained



(see examples in Fig. 3). In general, the LAP increased continuously with the concentrations of
MCA-leu, the AP stabilized around 1 µM MUF-P, whereas the βGLU showed intermediary
situations. The lowest activities were determined for βGLU (Table 2) and for this enzyme,
measurements were unfortunately below detection limits for most 'liw' and 'mdw' layers.
Occasionally, only few time series showed linear increases of fluorescence with time, coinciding
with the higher range of used concentrations. Consequently, fitting the Michaelis-Menten kinetic
was unfortunately impossible. The means of few βGLU rates measurable at depth were 0.010 ±
0.006 nmol L$^{-1}$ h$^{-1}$ in the 'liw' layer and 0.008 ± 0.006 nmol L$^{-1}$ h$^{-1}$ in the 'mdw' layer and could be
considered as a minimal value of Vm50.

Figures 4 to 6 show the distribution of Vm and Km for each station and in each layer sampled for
leucine aminopeptidase (LAP), β−glucosidase (βGLU) and alkaline phosphatase (AP). Activities
exhibited a large longitudinal variability, particularly AP, even at depth (CV ranged from 101 to
163% according layers, Fig. 4). The lowest longitudinal variability was obtained for βGLU within
surface layers (34% for Vm50, 45% for Vm1).

In all layers, the highest mean Vm of the 10 stations were obtained for AP, followed by LAP and
then βGLU, whatever the range of tested concentrations (Vm50 or Vm1, Table 2). Within 'surf' AP
Vm50 and AP Vm1 means were 4 and 6 times higher than their corresponding LAP means,
respectively. Within 'dcm' layers, these values were 6 and 10 times higher, respectively. Vm50 AP
average was 7 times higher than its corresponding βGLU average within 'surf' (and 5 times at
'dcm') and this factor increased considering Vm1 (16 and 20 times, respectively at 'surf' and 'dcm'
layers). For each enzyme, the order of magnitude reached for Vmax was the same at the 'surf' and
'dcm' layers.

For LAP (Fig. 4), Vm50 was on average 3 times higher than Vm1 in 'surf' and 'dcm' layers, but the
differences between these two rates increased with depth (x9 in 'liw', x12 in 'mdw'). Vm50
decreased from epipelagic to mesopelagic waters by a factor of 8 on average, (ratio 'depth variation
factor' – DVF), but by a factor x19 for Vm1 (Fig. 4a). However, if this decrease was particularly
obvious both for Vm1 and Vm50 for stations ST10 to ST5 in the Western Basin, it was not the case
for Tyrrhenian waters (ST5, TYR and ST6) where Vm1 decreased but not Vm50. Within ION, Vm1
decreased faster than Vm50. Km50 of LAP showed inconsistent patterns with depth. Some stations
showed a decrease at 'dcm' compared to 'surf' layers (SD10, SD2, TYR, SD6, ION). Km50 within
'liw' layers were on the same order of magnitude than at the surface, sometimes even higher
(FAST, ST 3, ST5, ST6, ION) as well as in the 'mdw' layers particularly in Tyrrhenian and Ionian
seas (Fig. 4b). For Km1, the trend was a decrease with depth in the western stations (ST10 to ST3)
whereas from stations 4 to ION the order of magnitude of Km1 at all depths were the same, with
sometimes a decrease within 'liw' layers (in Tyrrhenian stations).

For βGLU (Fig. 5), Vm50 was on average 7 and 5 times higher than Vm1 in 'surf' and 'dcm'
layers, respectively. Their differences in Vm were greater than those observed for LAP or AP (Fig.
5a). Km50 was of the same order of magnitude at 'surf' and 'dcm' layers or slightly lower (ST1,
ST2, ST3, ST6), but the opposite trend was observed for Km1, which tended to be equal or higher
within 'dcm' layer (Fig. 5b), in particular in the western stations (ST10, FAST, ST2, ST3).

AP was the enzyme for which Vm1 and Vm50 were the closest (Fig. 6a), with saturation rates
occurring already around 1 µM of added MUF-P (Fig. 3). Vmax50 to Vmax1 ratios were 1.6, 1.3,
2.4 and 3.0 on average for 'surf', 'dcm', 'liw' and 'mdw' layers, respectively. AP within 'surf' layer



showed a larger range of longitudinal variability than the remaining studied ectoenzymes. For instance, Vm50 ranged from 0.3 to 8.3 nmol L$^{-1}$ h$^{-1}$for AP, compared to 0.08 - 0.23 nmol L$^{-1}$ h$^{-1}$ for βGLU and 0.36 - 2.85 nmol L$^{-1}$ h$^{-1}$ for LAP (Table 2). The trend within 'surf' was an increase of AP

towards the east, from a range of 0.8 - 0.9 nmol L$^{-1}$ h$^{-1}$ for Vm50 at ST10 and FAST and up to 8 nmol L$^{-1}$ h$^{-1}$ at ION. Both AP Vm1 and Vm50 decreased with depth (Fig. 6a), although sometimes both AP Vm50 and AP Vm1 at the 'dcm' layer were higher than the surface (ST1, 2, 5 TYR, ION). At all stations Vm in 'mdw' were equal or lower than those within 'liw' layers. About the decline trend with depths, and in opposition with what was described for LAP, we could not see any

difference between the eastern and the western stations. DVF was large, varying from x2.8 to x71 for Vm50, with lower decreases with depths at ST10 (x2.8) FAST (x3.2) and ST3 (x2.4), and highest DVF at ST1 (x34), ST2 (x71) and ION (x54). Vm1 and Vm50 were almost equal (averages of Vm50/Vm1 ratio for the whole data set was 1.6 ± 0.5), and although for AP Km50 was on average 7 times higher than Km1, their differences were the lowest compared to the two other

enzymes. Average Km50/Km1 ratio for βGLU was 320 and average Km1/Km50 ratio for LAP was 118. Km1 and Km50 increased mostly with depth (Fig. 6b) except at ST10. For the other stations, the trend was that Km50 increased more with depth (DVF ranged from x2 to x29, seven stations) than Km1 (DVF ranged from x1.9 to x3.8, nine stations).

The turnover time of ectoenzymes was determined as the Km/Vm ratio, which drives the activity at

low concentrations of substrates. The incidence of the tested set of substrate concentration is very important on this parameter, as turnover times are systematically lower for the 25-1000 nM substrate range of concentrations (Table 4). The turnover times were the lowest for AP and the highest for βGLU. We estimated the degree of difference between the two-kinetics using the 'biphasic indicator' as developed in Tholosan et al. (1999). This index tracks the difference between

the initial slopes (Vm/Km) of Michaelis-Menten kinetics as (Vm1/Km1) / (Vm50/Km50). The biphasic indicator was particularly marked for βGLU (68 in surface and 29 at the dcm), for LAP it increased from about ~4 in 'surf 'and 'dcm' to 10 within 'liw' and 20 within 'mdw'. For AP it stayed more or less constant at the four layers sampled (range 1.9-3.4).

### 3.4 Specific activities

Both BP and BA were used to compute specific activities (Table 2, Fig S2). Bulk heterotrophic prokaryotic production (BP) was of the same order of magnitude within 'surf' and 'dcm' layers and decreased by a factor of 59 ± 23 on average within 'liw' and 'mdw' layers. Per layer, BA was less variable than ectoenzymatic activity or BP. The mean value at epipelagic layers decreased by a factor of 5 in 'liw' and by a factor of 10 in 'mdw' layer.

For LAP, specific activities per bacterial cell ranged from 0.1 - 2.1 x 10$^{-18}$ to 0.7 - 13 x 10$^{-18}$ mol leu cell$^{-1}$ h$^{-1}$, based on Vm1 and Vm50 rates, respectively (Fig. 7 a, b; Table 3 for Vm1). A significant decrease with depth between epipelagic waters and deep waters was obtained only for cell-specific Vm1 LAP, but not for cell-specific Vm50 LAP (p < 0.001). For AP, per cell-specific activities ranged from 0.11 to 32 x 10$^{-18}$ mol P cell$^{-1}$ h$^{-1}$and from 0.14 to 39 x 10$^{-18}$ mol P cell$^{-1}$ h$^{-1}$ based on

Vm1 and Vm50 rates, respectively, not differing significantly due to the low differences between APVm1 and APVm50 (Fig. 7 a, b). Cell-specific AP exhibited either an increase or a decrease with depth (Fig. 8). The depth variation factor (DVF) ranged from x0.1 to x28 (Table 3). Although the t-test gave no significant differences between epipelagic and deep layers for both specific activities, the medians of Vm1AP per cell for 'liw' and 'mdw' layers were lower than those of 'surf' and

'dcm' (Fig.7 c, d). Specific prokaryotic production ranged from 1 to 136 x 10$^{-18}$ g C bact$^{-1}$ h$^{-1}$



(Table 3) and always decreased with depth (DVF ranged x4-x23). If only LAP Vm1 rates are considered, while the specific activities per unit cell decreased with depth, the activity per unit BP increased with depth at all stations (Table 3). APVm1 per unit BP tended less to increase with depth than that per unit cell, although a large range of DVF was obtained in both cases according stations

for these parameters (Table 3). For all variables (the three enzymes and BP), the trend of specific activities with depth was highly variable among the different stations (Fig. 9). The highest DVF was obtained for BP per cell or AP per cell. Only cell-specific Vm 50 LAP, cell-specific Vm1 AP and cell-specific Vm50 AP sometimes increased with depth.

### 3.5 In situ hydrolysis rates

The *in situ* hydrolysis rates of TAA by LAP are higher using Vm1 and Km1 constants than using Km50 and Vm50, by ~3 times in epipelagic but ~7 times in deep waters (Fig. 9). Km50 were much higher than TAA concentrations (30 to 400 fold according layers). This difference was still visible but highly lowered considering Km1, as values between TAA and Km1differed by factor 2 to 5. Consequently, *in situ* TAA hydrolysis rates by LAP based on Km50 and Vm50 represented few

percent of Vm50 (means of 13% at 'dcm' to 0.6% at 'mdw'). However, based on Km1 and Vm1, *in situ* rates were relatively higher but in constant proportion relative to Vm1 (30 to 39% according layers).

The *in situ* hydrolysis rate of TCHO by βGLU are higher using Vm1 and Km1 constants than using Km50 and Vm50, by ~2.5 times in epipelagic (Fig. 10). Km50 were higher than TCHO

concentrations (Table 2, Table S1), by a mean factor of 18 ± 12 within 'surf' and 24 ± 18 at 'dcm' layers. Consequently, *in situ* βGLU hydrolysis rates based on Km50 and Vm50 were quasi proportional to the turnover rate Vm1/Km1and represented a mean of 7% of the Vm50 in epipelagic layers. At the opposite, Km1 were much lower than TCHO concentrations (by a factor 31 ±19 in 'surf', 8 ± 7 at the dcm) and thus most *in situ* rates based on Km1 and Vm1 were close to Vm1

(93% in 'surf', 79% at the 'dcm').

### 4. Discussion

### 4.1 The use of a broader set of substrate concentrations will change our interpretation of ectoenzyme kinetics

The idea that ectoenzyme kinetics are not monophasic is nor new nor surprising (Sinsabaugh and Shah, 2012 and references therein). But despite the 'sea of gradients' encountered by marine bacteria (Stocker, 2012), multiphasic kinetic is seldom considered. In this work, we compare the low and high concentration ranges of fluorogenic substrates in order to evaluate the possibility of considering the different kinetic parameters in relation to the *in situ* natural concentrations of the

substrates. Unanue et al. (1999), in the coastal, epipelagic Mediterranean Sea in winter, used a set of concentration from 1 nM to 500 µM, revealing biphasic kinetics with a switch between the two phases around 10 µM for LAP and 1 - 25 µM for βGLU. They referred to 'low affinity systems' and 'high affinity systems. They also observed that the differences between the Vm of the two types of enzymes varied according to the range of substrates analyzed. Bogé et al. (2012) used a MUF-P

range from 0.03 to 30 µM and described biphasic AP kinetics in the Toulon Bay, with a switch between the 2 enzymatic systems around 0.4 µM. The biphastic factor as defined in Tholosan et al (1999) helps to determine the degree of difference between the two Michaelis-Menten kinetics. In our data set, it was 4.5, 3.7, 11 and 20 for LAP in 'surf', 'dcm', 'liw' and 'mdw' layers,



respectively, showing enhanced differences in mesopelagic compared to epipelagic waters. Thus,
the differences between these two LAP enzymatic systems could reach differences in the water
column as much as in sediments (Tholosan et al., 1999), in which large gradients of organic matter
lability take place. However, it was not the case for all enzymes, as the differences were the lowest
for AP (biphasic factor 3.3, 1.9, 2.4 and 1.7 in 'surf', 'dcm', 'liw' and 'mdw' layers, respectively)
and relatively consistent with depth. Finally, the difference between the 2 types of enzymes was
greater for βGLU (biphasic factor 68 for 'surf' and 29 for 'dcm' layers).

In this study, the use of MUF-P concentrations ranging between 0.025 and 1 µM highlighted that
AP rates fit well with the Michalis-Menten Kinetic model, with saturations reached at 1 µM. This
AP activity, responding to a low concentration range, should belong to free-living bacteria and/or
dissolved enzymes (< 0.2 µm fraction) with affinities adapted to low concentrations of substrates.
These results are in agreement with average DOP concentration measured, ranging between 12 and
122 nM in epipelagic waters (Pulido-Villena et al., this issue in prep) and, when detectable,~ 40 nM
in deep layers. Using fractionation-filtration procedures, much recent works have shown that more
than 50% of the AP activity could be measured in the < 0.2 µm size fraction (Baltar, 2018 and
references therein),whereas the dissolved fraction of other enzymes is generally lower. Hoppe and
Ulrich (1999) found a contribution of the < 0.2 µm fraction of 41% for AP, 22% for LAP and only
10% for βGLU. We tried during the PEACETIME cruise few size fractionation experiments on
some occasions in 'surf' and 'dcm' layers (results not shown). Although the size-fraction < 0.6 µm
was filtered only by gravity and a gentle vacuum was applied for the < 0.2 one, AP in the < 0.2 µm
fraction exceeded the total in 3 cases out of 12. Setting these cases to 100%, we obtained a mean
contribution of 60 ± 34% (n = 12) for this size-fraction. However, contributions were on average 25
± 16% (n = 12) for βGLU and 41 ± 16% (n = 12) for LAP for this same size-fraction (< 0.2 µm),
confirming these trends in the Mediterranean Sea. At the opposite, the higher differences in LAP
Vm and Km, in response to the range of tested concentrations, suggest development of activities
facing large gradients of substrate concentrations.

The use of a large range of concentrations  impacted also the Km values. Indeed, excluding low
concentrations has consequences on the Km estimation, because if only a high concentration range
is used, the kinetic contribution of any system with high affinity is hidden. Baltar et al. (2009b),
using a concentration of substrates ranging from 0.6 to 1200 µM, reported an increase in the LAP
Km (~400 to 1200 µM) and AP Km (~2 to 23 µM) with depth down to 4500 m in the sub-tropical
Atlantic. In contrast, Tamburini et al. (2002), using a concentration of substrates ranging from 0.05
to 50 µM, obtained lower Km values (ranging between 0.4 and 1.1 µM) for LAP in the
Mediterranean deep waters (down to 2000 m depth). It is however difficult to conclude about the
effect of the substrate concentration on Km variability with depth by comparing 2 studies from
different environments. In our study where both kinetics were determined in the same waters, we
came to different conclusions considering Km50 or Km1: if LAP Km1is on average x 24 (surf) to x
8 (dcm) higher than βGLU Km1, Km50 were in the same order of magnitude: 7 vs 10 µM in 'surf'
and 5 vs 8 µM in 'dcm' layers for LAP and βGLU, respectively. On the other hand, ratios of
enzymatic properties are also relevant information for the interpretation of the ectoenzymatic
hydrolysis of the substrate in terms of quality and quantity. For instance, using high concentration
ranges, the LAP Km is largely higher than βGLU Km probably because LAP is not adapted to face
low concentration ranges, in contrast to βGLU (Christian and Karl, 1995). However,  when the
fluorogenic substrates have the same range of concentration as the natural substrates used by the
enzyme targeted with the fluorogenic substrate, competition for the active sites could be possible.





This increases the risk of overestimating Km1. We thus assumed that Km1, although lower than in
published values, are still potentially overestimated. Another difference in response to the tested
range of concentrations for each substrate was the turnover time (Km/Vm ratio): the lower the
Km/Vm, the better the adaptation is to hydrolyze substrates at low concentrations. This should be
considered carefully when comparing reported values.

### 4.2 Trends with depth

We have shown that, depending on the range of concentrations tested, different conclusions can be
drawn regarding the debate about the increase or at least maintenance of specific levels of activity
within deep layers (Koike and Nagata, 1997; Hoppe and Ulrich, 1999; Baltar et al., 2009b).The
trend in vertical depth of activities and cell-specific activities will be discussed below in the context
of the use of different concentration sets for kinetics.

In order to compare enzymatic activities between different water masses, from epi- to meso- and
bathypelagic layers, alongside to the decrease with depth of bulk activities, the use of a normalized
activity is required. As most enzymes are produced by heterotrophic prokaryotes, normalization
over total heterotrophic prokaryotic cells is common. However, it is recognized that heterogeneous
distribution of single-cell activities among heterotrophic prokaryotes is a recurrent bias in
interpretation of this parameter (Martinez et al., 1996; Hoppe and Ulrich, 1999; Baltar et al.,
2009b). In addition, within epipelagic waters some phytoplankton species also express
ectoenzymatic activity such as AP (Dyhrman and Palenik, 1999, Sebastian and Ammermann 2009,
Lin et al., 2012) and LAP (Martinez and Azam, 1993). At depth, gravitational sinking particles
could retain phytoplankton flocs still showing physiological capacities (Lochte and Turley, 1988;
Agusti et al., 2015).

Increasing AP activities per cell with depth has been reported in the Indian Ocean (down to 3000 m-
depth; Hoppe and Ullrich, 1999), in the subtropical Atlantic Ocean (down to 4500 m-depth; Baltar
et al.; 2009b) in the central Pacific Ocean (down to 4000 m-depth; Koike and Nagata,1997). These
authors used high concentrations of MUF-P (250 μM, concentration kinetics from 0.6 to 1200 μM
and 150 μM, for these authors, respectively) that could stimulate ectoenzymes of cells attached on
suspended or sinking particles, and thus adapted to face higher concentration ranges. However,
these trends were also obtained using low concentrations (max 5 μM MUF-P), at depths down to
3500 m in the Tyrrhenian Sea (Tamburini et al., 2009). In the bathypelagic layers of the central
Pacific, AP rates accounted for as much as half of that observed in the epipelagic layer but the < 0.2
560 μm dissolved AP was removed to estimate bathypelagic activities (Koike and Nagata, 1997). They
suggested that the deep-sea AP is due to fragmentation and dissolution of rapidly sinking particles.
Indeed, it has been shown that AP determined on concentrated particles had the highest
concentration factor compared to AP of bulk seawater among different tested enzymes (Smith et al.,
1992). Note, however, that our data generally stops in mesopelagic layers (1000 m). Tamburini et
565 al. (2002) obtained a different relative contribution of deep-sea samples when used MUF-P
concentrations were 25 nM or 5 μM at the DYFAMED station in the NW Mediterranean Sea (down
to 2000 m-depth), further showing the artifact of the concentration used. Furthermore, as shown by
these authors, the deep activities could be x1.4 to x2.6 times higher due to the effect of hydrostatic
pressure out of convective periods. With concentration kinetics ending at 50 μM of MUF-P, the
570 specific activities of AP reached using per cell Vm50 or per cell Vm1 were not so different and
their trend with depth were similar (Fig. 8). The particulate matter C/P ratio did not change with



depth whereas DOC/DOP ratio decreased (from 2200-2400 to 1500-1200), suggesting a preference for heterotrophic prokaryotes to use dissolved organic phosphorus as substrate of AP.

For LAP activity, however, Vm50 decreased with depth more intensively than Vm1, but cell-specific LAP showed contradictory results: at all stations cell-specific Vm1 decreased with depth (according to the DVF criterion, Fig. 8) whereas Vm50 remained stable (2 stations over 10) or increased with depth (5 stations over 10).Using a high concentration of MCA-leu other authors have systematically found an increase in LAP activity per cell with depth in bathypelagic layers (Zaccone et al., 2012; Caruso et al., 2013). From our data set, among the two parameters LAP Vm and LAP Km, it is LAP Km which showed the largest differences between the 2 types of kinetics. Km 50 particularly increased with depth more than Km1, and the ratio Km50/Km1 switched from ~16 in epipelagic waters to 121 and 316 in 'liw' and 'mdw' layers, respectively. At many stations (TYR, ION, FAST and ST10), LAP Km1 was stable or decreased with depth whereas LAP Km50 increased, suggesting that within deep layers the LAP activity was more linked to the availability of suspended particles or fresh organic matter associated to sinking material, than to DON. Thus, the difference between Km1 and Km50 might reflect a strategy to adapt to a potential spatial and/or temporal patchiness in the distribution of suspended particles. Freshly sinking material is statistically not included in the bulk, because of the small volume of water incubated, but could contribute to the release of free bacteria, small suspended particles and DOM within its associated plume (Azam and Long, 2001; Tamburini et al., 2003; Grossart et al., 2007; Fang et al., 2014). Baltar et al. (2009a) also suggested that hot spots of activity at depth were associated with particles. The fact that the C/N ratio of particulate material increased (from 11-12 to 22-25) but not that much for DOC/DON (13-12 to 14-15 from 'surf' and 'dcm' to 'liw' and 'mdw', respectively) confirms a preferential utilization of proteins substrates from particles. Recently, Zhao et al. (2020), based on the increasing contribution of genes encoding secretory enzymes, suggested that deep-sea prokaryotes and their metabolism are likely associated with particles rather than on the utilization of ambient-water DOC.

Variation in the relative activities of different enzymes is suggested as a possible indicator of changes of bacterioplankton nutrition patterns along the water column. The LAP/βGLU ratio decrease with depth follows the decrease of protein to carbohydrate ratio of particulate material (Misic et al., 2002), nitrogen being remineralized faster than carbon. However, here, the TCHO-C/TAA-C ratio was consistently higher within the 'dcm' layer (~90 m) than at the surface and the/ LAP/βGLU ratio generally increased also as a consequence. Below the 'dcm' layer, we estimated Vm50 LAP/βGLU ratios from few of the single rates measured at high concentration (above LOD), and observed an increase with depth. Despite the particulate C/N ratio increasing with depth and TCHO-C/TAA-C increasing in the 'mdw' layer, LAP increased faster than βGLU with depth. Others have also shown an increase of LAP/βGLU ratio with depth (Hoppe and Ullrich, 1999 in Indian Ocean, Placenti et al., 2018 in the Ionian Sea). Many factors, such as the freshness of the suspended particles, a recent event of convection, a lateral advection from margins, as well as the seasonality and taxonomic composition of phytoplankton could influence dynamics at depth, particularly in the mesopelagic layer (Severin et al. 2016, Caruso et al 2013).

### 4.3 Regional variability

In epipelagic waters, both AP maximum rates (Vm1, Vm50) significantly increased from the Algerian/Ligurian Basins to the Tyrrhenian Basin (t test, p = 0.002 and p = 0.02, respectively), and reached maximum values at ION. This longitudinal increase in AP activity was also confirmed by





calculating specific activities which also increased towards ION. This increase of cell-specific AP appears to follow a decrease in DIP availability. While DIP can be assimilated directly through a high affinity absorption pathway, the assimilation of DOP requires that the molecule is first remineralized to separate the DIP from the carbon fraction, using AP enzymes. POP is an indicator

of living biomass and enzyme producers, but the correlation between VmAP and POP were negative in the surface layers (log-log relationship, r = - 0.86, -0.88 for Vm50 and Vm1, respectively), suggesting that POP reflected more the progressive eastward decline of living biomass and its increased capacity to derepress AP genes. VmAP rates in the surface did not correlate with DIP, however the relative DIP deficiency increased eastward, suggested by the

deepening of the phosphacline (Table 1), the decrease of average DIP concentration within the phosphate-depleted layer and the decrease in P diffusive fluxes reaching the surface layer (Pulido-Villena et al. 2020, in prep, this issue). Along a trans-Mediterranean transect, Zaccone et al. (2012), did not get any trend between DIP and AP as well, although they found also increased values of AP specific activities in the Eastern Mediterranean Sea. Bogé et al. (2012), using a concentration set

close to ours (0.03-30 μM MUF-P) obtained differences in Vm for the 2 types of kinetics (contrary to our results) and described different relationships with DOP and DIP according to the low and high affinity systems. Such differences could be due to the large gradient of trophic conditions in their study, involving a eutrophic bay where DOP and DIP concentration ranged from 0 to 185 nM, and from 0 to 329 nM, respectively. In order to circumvent the effect of depth, correlations are

described in our study only for 10 surface data where DIP concentration range is narrow (4-17 nM).

The AP/LAP ratio can be used as an indicator of N - P imbalance as demonstrated in enrichment experiments (Sala et al. 2001). In this study, large concentrations of substrates were used (200 μM) and they described a decrease of the AP/LAP ratio after DIP addition and conversely, a large increase of it (10 fold) after addition of 1 μM nitrate. In their initial experimental conditions, the

ratio ranged from 0.2 to 1.9. We observed a similar low ratio in the western Mediterranean Sea, but in the Ionian Sea the AP/LAP reached 17 (Vm50) and 43 (Vm1) suggesting that nutrient stress and imbalance can be as important and variable in different regions of the Mediterranean Sea as observed after manipulation of nutrients.

LAP/βGLU ratio is used as an index of the ability of marine bacteria to preferentially metabolize

proteins rather than polysaccharides. The prevalence of LAP over βGLU is a recurrent observation in temperate areas (Christian and Karl, 1995; Rath et al., 1993) and in high latitudes (Misic et al., 2002, Piontek et al., 2014). For example, LAP/βGLU ratio varied widely from the Equator to the Southern Ocean, with values from 0.28-to 593 (Sinsabaugh and Shah, 2012). In the Ross Sea, this ratio exhibited a relationship with primary production (Misic et al., 2002). In the Caribbean Sea,

along an eutrophic to oligotrophic gradient, the LAP/βGLU ratio increased toward oligotrophy (Rath and Herndl, 1993). In epipelagic zone, during our study, the degree of trophic conditions exhibited a small gradient of productivity (18 to 35 mg Chla m$^{-2}$) along the western to the eastern Mediterranean Sea. Following this gradient, LAP /βGLU ratio ranged from 3 to 17 for Vm50, and from 8 to 34 for Vm1 and thus varied according to the concentration range tested. However we

found no statistical significant difference between Tyrrhenian Sea and the western area, due to a high intra-regional variability. Unanue et al. (1999) established biphasic kinetics and obtained a ratio VmLAP / VmβGLU around 20 for their low concentration range and around 10 for their high concentration range, confirming our results. Finally, the LAP/βGLU ratios reported in this study and other works using low substrates ranges are still lower than when using higher concentrations:

20-200 in the subarctic Pacific (Fukuda et al., 1995, using 200 μM concentration), 213 at station





ALOHA in the equatorial Pacific (Christian and Karl, 1995, using LLBN instead of MCA-leu at 1000 µM and MUF-βGLU at 1.6 µM), suggesting that the ratio LAP/βGLU highly changes according to the fluorogenic substrate concentration and not in a regular way.

Within deeper layers, cell-specific LAP and cell-specific AP tended to be higher within Tyrrhenian
Sea and ION stations than on the westernmost stations, although comparisons between both areas were insignificant due to the high intra-regional variability. Azzaro et al. (2012) showed high seasonal variability in heterotrophic metabolism in the Southern Adriatic Pit, down to 1200 m depth, and they relied this variability to deep convection events. Tamburini et al. (2002, 2009) noticed also a seasonal variability on deep samples depending on seasonal variability of surface
productivity and particle fluxes. During an intense water column mixing (0-1500 m) in the NW Mediterranean Sea, Severin et al. (2016) showed that deep-sea convection enhanced bacterial abundance and bacterial heterotrophic production at depth, resulting in a drastic decline of cell-specific extracellular enzymatic activities (up to 67%).

**4.4 Potential contribution of macromolecules hydrolysis to bacterial production**

Computation of *in situ* hydrolysis rates demonstrated the direct influence of the determination of Km and Vm using an appropriate set of fluorogenic concentration to compute rates. TAA concentrations were lower than Km1 and Km50. The two Michaelis–Menten plots cross each other, at a substrate value of about 1.8 ± 1.3 µM with LAP and 1.7 ± 0.6 µM with βGLU. Considering the TAA range, the high affinity system (Km1 Vm1) with its low Km and higher turnover rates gave
consequently better *in situ* rates than the low affinity system (Km50 Vm50). Although TCHO ranges were lower than Km1 but higher than Km50, TCHO was always lower than the crossing concentration point of the two types of kinetics, and consequently, again, the high affinity system gave higher *in situ* hydrolysis rates than the low affinity system. If the experimentally added substrate concentration is clearly above the possible range of concentrations found in the natural
environment, *in situ* rates could be largely overestimated. To obtain a significant determination of the *in situ* rates, the added substrate concentrations should be close to the range of variation expected in the studied environment (Tamburini et al., 2002).

We compared the *in situ* LAP hydrolysis rates to the N demand of heterotrophic prokaryotes (which was based on bacterial production data assuming C/N ratio of 5 and no active excretion of
nitrogen), and the *in situ* rates of TAA plus TCHO to the bacterial carbon demand (based on a bacterial growth efficiency of 10% (Gazeau et al, this issue, in prep, Céa et al., 2014, Lemée et al., 2012). Using the low affinity constants (Vm50 and Km50), hydrolysis of TAA by LAP contributed only to 25% ± 22% of the bacterial N demand in epipelagic layers and 26% ± 24% in deep layers. This contribution increased using the high affinity system (48% ± 29% and 180% ± 154% in
epipelagic layers and deep layers, respectively). In the North Atlantic, LAP hydrolysis rates of particles (0.3 µM MCA-leucine added) to bacterial nitrogen demand varied between 63 and 87%, increasing at 200 m. Crottereau and Delmas (1998) combined kinetics of LAP with combined amino acid concentrations and found a range of 6 – 121% contribution to bacterial N demand in aquatic eutrophic ponds. A large variability of LAP hydrolysis to bacterial N demand has been also
detected in coastal-estuarine environments using a radiolabeled natural protein as a substrate (2 – 44%, Keil and Kirchman, 1993). Pointek et al. (2014), along a 79°N transect in North Atlantic, used the turnover of βGLU and LAP determined with 1 µM analog substrate concentrations to compute *in situ* TCAA and TCHO hydrolysis rates and showed that 134% and 52% of BP could be supported by substrates issued from peptide and polysaccharides activities, respectively. Based on a





BGE of 10%, these fluxes will represent 10 times less, i.e. 13 and 5% of bacterial carbon demand,
which is in the same order of magnitude as we obtained. In our study, the contribution of TAA
hydrolysis to bacterial N demand is increasing within the 'dcm' compared to the 'surf' layer (from
means of 10 to 40% based on the high affinity system). This is consistent, however, as some
cyanobacteria can express also LAP (Martinez and Azam, 1993) and *Synechococcus* and

*Prochlorococcus* are dominant phytoplankton groups in the Mediterranean Sea (Siokou-Frangou et
al., 2010). In our study, the 'dcm' was an active biomass layer where primary production (PP)
peaked (Maranon et al., 2020). Size fractionation of primary production showed the importance of
the phytoplankton excretion, contributing from 20 to 55% to total PP according the stations
(Maranon et al., 2020.). Within the surface mixed layer, other sources of N like atmospheric

deposition could sustain a significant part of bacterial N demand. The dry atmospheric deposition
(inorganic+ organic) of N at all the stations within the PEACETIME cruise corresponded to 25 ±
17% of bacterial N demand (Van Wambeke et al, in prep).

Likewise, the *in situ* cumulative hydrolysis rates of TCHO by βGLU, estimated only in epipelagic
layers were also ~3 times higher using the high affinity system. We summed C sources coming

from the hydrolysis by LAP and by βGLU in epipelagic layers (Fig. 10) and compared them to the
bacterial carbon demand. Dissolved proteins and combined carbohydrates contributed only to a
small fraction of the bacterial carbon demand: 1.5% based on the low affinity system constants and
3% based on the high affinity system.

It is only within deeper layers that the hydrolysis rates of TAA were at some stations more

important than bacterial N demand, suggesting that proteolysis is one of the major sources of N for
heterotrophic bacteria in aphotic layers. However, it was only based on Vm1 and Km1 kinetic
parameters (i.e. the high affinity system) that we found cases of over-hydrolysis of organic nitrogen
(Fig. 9). This over-hydrolysis was particularly marked in the LIW water mass of the Tyrrhenian
Basin, in which over-hydrolysis up to 220% was obtained as well as higher TAA concentrations in

comparison to "older' LIW waters in the Algerian Basin. TAA decreased faster than DON along
LIW trajectory so the labile DON fraction (combined aminoacids) was degraded first. The role of
sedimenting particles or large aggregates associated with attached bacteria are considered as major
providers of labile organic matter for free bacteria (Smith et al., 1992). We could consider that with
the 5 mL volume of water hydrolyzed for TAA analysis, and in the 2 mL water volume used to

determine ectoenzymatic kinetics, most of this particulate detrital pool of large size or high density
(i.e. fast sinking particles) is underrepresented,  thus the contribution of TAA hydrolysis to bacterial
nitrogen demand is underestimated. However, there is an increasing evidence of secretion not only
of monomers issued from hydrolysis, but also of ectoenzymes produced by particles-attached deep-
sea prokaryotes themselves (Zhao et al., 2020). This could explain why, in a small volume of bulk

sea water sample, not representative of large or fast sedimenting particles, we still found biphasic
kinetics. Studying alkaline phosphatase activity in the Toulon Bay, Bogé et al. (2013) observed
biphasic kinetics only in the dissolved phase, which also suggests that AP low affinity system
originates from enzyme secretion from particles. Afterwards, this team focused their research by
size fractionation of particulate material, and they found that the origin of the low affinity system

was mostly due to the > 90 µm fraction, i.e. large particles (Bogé et al., 2017).

**5 Conclusions**

The use of microplate titration technique improved greatly the simultaneous study of different
ectoenzymatic activities. Vertical and regional variability of activities were shown in the





Mediterranean Sea, where heterotrophic prokaryotes face not only carbon, but also N, P limitations.
Although biased by the use of artificial fluorogenic substrates, ectozenzymatic activity is an appropriate tool to study the adaptation of prokaryotes to the large gradients in stoichiometry, chemical characteristics and quantities of organic matter they face, especially when using a large panel of concentrations. Further combination of such techniques with the chemical identification of DOC and DON pools, and meta-omics, as well as the use of marine snow catchers, would help our
understanding of the biodegradation of organic matter in a 'sea of gradients'.

**Acknowledgements**: This study is a contribution of the PEACETIME project (http://peacetime-project.org), a joint initiative of the MERMEX and ChArMEx components supported by CNRS-INSU, IFREMER, CEA, and Météo-France as part of the program MISTRALS coordinated by INSU (doi: 10.17600/17000300). The authors thank also many scientists & engineers for their
assistance with sampling/analyses: J Ras for TCHO, R Flerus for TAA, J Guittoneau for nutrients, T Blasco for POC, J Uitz and C Dimier for TChl*a* (analysed at the SAPIGH HPLC analytical service at the IMEV, Villefranche), I Obernosterer, P Catala and B Marie for DOC and bacterial abundances. We warmly thank C Guieu and K Deboeufs, as coordinators of the program PEACETIME.

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





**Figure Legends**

Figure 1. Sampling sites. Color codes on dots correspond to the plots on Fig. 2.

Figure 2. Physical properties along the different sites: a) Oxygen versus depths, b T/S diagram.
Color codes correspond to the stations mapped Fig. 1. Principal water masses are indicated. MAW:
Modified Atlantic Waters, LIW: Levantine intermediate Waters, WMDW: Western Mediterranean
Deep waters, EMDW: Eastern Mediterranean Deep Waters.

Figure 3. Example of Michaelis-Menten plots, dcm layer at station FAST. Dots are data, continuous
lines the nonlinear regression plot derived from all concentrations added (0.025 to 50 µM). Smallest
graphs inside show dotted lines corresponding to regression plots derived from a restricted substrate
range (0.025-1 µM).

Figure 4. Distribution of leucine aminopeptidase (LAP) kinetic parameters Vm (a) and Km (b). Red
values are Vm50 and Km50 derived from the whole substrate MCA-leu range (0.025-50 µM) and
pink values are Vm1 and Km1 derived from a restricted substrate range (0.025-1 µM). The bars are
not cumulative but absolute values. The error bars are standard errors derived from the nonlinear
regressions. At each station four data are presented, corresponding to, from left to right, 'surf',
'dcm',' liw' and 'mdw' layers, respectively.

Figure 5. Distribution of βglucosidase (βGLU) kinetic parameters Vm (a) and Km (b). Yellow
values are Vm50 and Km50 derived from the whole substrate MCA-leu range (0.025-50 µM) and
white values are Vm1 and Km1 derived from a restricted substrate range (0.025-1 µM). The bars
are not cumulative but absolute values. The error bars are standard errors derived from the nonlinear
regressions. At each station four data are presented, corresponding to, from left to right, surface and
dcm layers, respectively. For 'liw' and 'mdw' layers (black bars) we presented only a mean
available rate detectable (kinetics were impossible to compute due to the low range of rates
measurable). This value is assumed to represent a minimal value for Vm50.

Figure 6. Distribution of alkaline phosphatase (AP) kinetic parameters Vm (a) and Km (b). Blue
values are Vm50 and Km50 derived from the whole substrate MUF-P range (0.025-50 µM) and
pale blue values are Vm1 and Km1 derived from a restricted substrate range (0.025-1 µM). The
bars are not cumulative but absolute values. The error bars are standard errors derived from the
nonlinear regressions. At each station four data are presented, corresponding to, from left to right,
'surf', 'dcm', 'liw' and 'mdw' layers, respectively. Vm1 and Km1 are not available at station FAST
mdw layer, due to a lack of significant rates below 1 µM MUF-P added.

Figure 7. Box plot distributions of specific Vm1 and Vm50 per bacterial cell, for alkaline
phosphatase (a: per cell Vm50AP, b: per cell Vm1AP) and leucine aminopeptidase (c: per cell
Vm50 LAP, d: per celVm1LAP). Box limits 25% and 75% percentiles, horizontal bar is median,
red cross is mean, blue dots are outliers.

Figure 8. Depth decreasing factor (DVF, unitless) among different specific activities. DVF is
calculated as the mean of pooled data from 'surf' and 'dcm' layers divided by the mean of pooled
data from 'liw' and 'mdw' layers. In red per cell Vm50LAP and per cell Vm1LAP: DVF of specific
aminopeptidase activities calculated for large (0.025-50 µM) and low (0.025-1 µM) substrate range,
respectively; in blue same thing for alkaline phosphatase (per cell Vm50AP, par cell Vm1AP). For
β−glucosidase, specific activities are based on the few detectable rates at high concentration (per
cell Vβglu, yellow dots). Black crosses are specific heterotrophic prokaryotic production per cell
(per cell BP).



Figure 9. In situ hydrolysis rates of dissolved proteins and particulate detrital N-proteins (nmol N L$^{-1}$ h$^{-1}$), determined from LAP ectoenzyme kinetics Vm1/Km1 versus Vm50 Km50, and comparison to heterotrophic bacterial nitrogen demand. a) epipelagic layers (surf, dcm), b) deeper layers (liw, mdw)

Figure 10. In situ rates of hydrolysis of dissolved and particulate detrital carbohydrates and C-proteins (nmol C L$^{-1}$ h$^{-1}$), determined from LAP and βGLU ectoenzymatic parameters Vm1 &Km1 versus Vm50 & Km50, and comparison to heterotrophic bacterial carbon demand (BCD) in epipelagic waters. Note the x10 scale for bacterial carbon demand on the right.





Table 1: Characteristics of the stations.

| | sampling date | Latitude °N | Longitude °E | bottom depth m | temp at 5 m °C | dcm depth m | Depth of 50 nM NO3 m | Depth of 50 nM DIP m | Integrated Chl a mg chla m$^{-2}$ | depth of 'ilw' layer sampled m | depth of 'mdw' water sampled m |
|---|---|---|---|---|---|---|---|---|---|---|---|
| ST 10 | 6/8/2017 | 37.45 | 1.57 | 2770 | 21.6 | 89 | 30 | 69 | 28.9 | 500 | 1000 |
| FAST | 6/3/2017 | 37.95 | 2.92 | 2775 | 21.0 | 87 | 50 | 59 | 27.3 | 350 | 2500 |
| ST 1 | 5/12/2017 | 41.89 | 6.33 | 1580 | 15.7 | 49 | 48 | 76 | 35.0 | 500 | 1000 |
| ST 2 | 5/13/2017 | 40.51 | 6.73 | 2830 | 17.0 | 65 | 40 | 70 | 32.7 | 500 | 1000 |
| ST 3 | 5/14/2017 | 39.13 | 7.68 | 1404 | 14.3 | 83 | 47 | 100 | 23.2 | 450 | 1000 |
| ST 4 | 5/15/2017 | 37.98 | 7.98 | 2770 | 19.0 | 64 | 42 | 63 | 29.2 | 500 | 1000 |
| ST 5 | 5/16/2017 | 38.95 | 11.02 | 2366 | 19.5 | 77 | 42 | 78 | 30.5 | 200 | 1000 |
| TYR | 5/17/2017 | 39.34 | 12.59 | 3395 | 19.6 | 73 | 82 | 95 | 31.3 | 200 | 1000 |
| ST 6 | 5/22/2017 | 38.81 | 14.50 | 2275 | 20.0 | 75 | 43 | 113 | 18.7 | 400 | 1000 |
| ION | 5/25/2017 | 35.49 | 19.78 | 3054 | 20.6 | 105 | 85 | 231 | 27.7 | 250 | 3000 |





Table 2. Microbial abundances and fluxes at the 4 layers studied for ectoenzymatic activities. Means ± sd and range values given for all stations (n=10), and both range of concentrations tested (up to 50 or up to 1 µM). Maximum velocity rates (Vm50 and Vm1), half saturation constants (Km50 and Km1) for leucine aminopeptidase (LAP), β-glucosidase (βGLU), alkaline phosphatase (AP), bacterial abundance (BA) and heterotrophic prokaryotic production (BP). ld limits of detection, not enough data to plot Michaelis-Menten kinetics

| | | surface | dcm | liwlayers | mdw waters |
|---|---|---|---|---|---|
| Vm50 LAP | mean ± sd | 1.00 ± 0.78 | 1.20 ± 0.92 | 0.26 ± 0.24 | 0.17 ± 0.13 |
| nmol $l^{-1}$ $h^{-1}$ | range | 0.36 – 2.85 | 0.33 – 2.83 | 0.08 – 0.91 | 0.06 – 0.45 |
| Vm1 LAP | mean ± sd | 0.29 ± 0.10 | 0.45 ± 0.25 | 0.028 ± 0.014 | 0.017 ± 0.010 |
| nmol $l^{-1}$ $h^{-1}$ | range | 0.21 – 0.56 | 0.19 – 0.98 | 0.014 – 0.060 | 0.007 – 0.042 |
| Vm50 βGLU | mean ± sd | 0.13 ± 0.04 | 0.12 ± 0.08 | ld | ld |
| nmol $l^{-1}$ $h^{-1}$ | range | 0.08 – 0.23 | 0.03 – 0.30 | | |
| Vm1 βGLU | mean ± sd | 0.019 ± 0.009 | 0.025 ± 0.019 | ld | ld |
| nmol $l^{-1}$ $h^{-1}$ | range | 0.012 – 0.040 | 0.014 – 0.077 | | |
| Vm50 AP | mean ± sd | 2.56 ± 2.58 | 3.73 ± 4.52 | 0.38 ± 0.48 | 0.25± 0.40 |
| nmol $l^{-1}$ $h^{-1}$ | range | 0.30 – 8.30 | 0.11– 14.6 | 0.04 – 1.66 | 0.06 – 1.30 |
| Vm1 AP | mean ± sd | 1.55 ± 1.58 | 3.01 ± 4.01 | 0.02 ± 1.11 | 0.12 ± 0.25 |
| nmol $l^{-1}$ $h^{-1}$ | range | 0.25–5.62 | 0.07–13.2 | 0.24 – 0.33 | 0.01 – 0.80 |
| Km50 LAP | mean ± sd | 7.4 ± 6.9 | 5.2 ± 7.7 | 17.8 ± 15.2 | 22.3 ± 26.3 |
| µM | range | 0.8–20.9 | 0.4–25.0 | 3.6 – 41.6 | 1.8 – 83.8 |
| Km1 LAP | mean ± sd | 0.50 ± 0.20 | 0.42 ± 0.28 | 0.23± 0.19 | 0.25± 0.27 |
| µM | range | 0.12–0.83 | 0.07–0.90 | 0.10 – 0.69 | 0.01 – 0.88 |
| Km50 βGLU | mean ± sd | 10.6 ± 6.3 | 8.7 ± 7.2 | ld | ld |
| µM | range | 4.4–27.4 | 1.2–24.3 | | |
| Km1 βGLU | mean ± sd | 0.044 ± 0.071 | 0.11±0.11 | ld | ld |
| µM | range | 0.009–0.244 | 0.01 – 0.36 | | |
| Km50 AP | mean ± sd | 0.72 ± 0.71 | 0.49 ± 0.34 | 2.25 ± 2.42 | 3.7 ±6.8 |
| µM | range | 0.09–2.18 | 0.18–1.07 | 0.17 – 7.32 | 0.4 – 21.9 |
| Km1 AP | mean ± sd | 0.11 ± 0.03 | 0.27 ± 0.28 | 0.37 ± 0.22 | 0.27 ± 0.16 |
| µM | range | 0.07–0.14 | 0.05 – 0.80 | 0.14 – 0.89 | 0.06 – 0.52 |
| BA | mean ± sd | 5.3 ± 1.6 | 5.4 ±1.5 | 1.13 ± 0.40 | 0.56 ± 0.15 |
| $10^5$ cells $ml^{-1}$ | range | 2.1–7.8 | 4.0 – 8.5 | 0.41 – 1.91 | 0.33 – 0.78 |
| BP | mean ± sd | 37 ± 13 | 21 ± 7 | 0.77 ± 0.40 | 0.27 ± 0.19 |
| ng C $l^{-1}$ $h^{-1}$ | range | 26 – 64 | 12 – 32 | 0.39 – 1.60 | 0.07 – 0.60 |

Table 3 Range of different potential specific activities calculated using Vm derived from the low range of concentration tested (25 to 1000 nM fluorogenic substrates; Vm1), and specific to either i) abundance of total heterotrophic prokaryotes (per cell activities), ii) heterotrophic bacterial production (per unit BP)and iii) particulate organic matter: nitrogen (PON) for LAP, carbon (POC) for βGLU and phosphorus (POP) for AP. DVFis the 'depth decreasing factor', calculated for each station as mean value in epipelagic water (surface and dcm data) divided by the mean in deep waters (low and mdw data) . The distribution of cell specific Vm1 and cell specific Vm50 for AP and LAP are also presented on Fig 7.

| enzyme | units | surface | dcm | liw | mdw | DVF |
|---|---|---|---|---|---|---|
| Per cell LAP | $10^{-18}$ mol leu bact$^{-1}$h$^{-1}$ | 0.33-1.52 | 0.44-2.18 | 0.11-0.70 | 0.12-0.54 | *1.3-9.6* |
| Per cellβGLU | $10^{-18}$ mol glucose bact$^{-1}$ h$^{-1}$ | 0.02-0.11 | 0.02-0.17 | nd | nd | *nd* |
| Per cell AP | $10^{-18}$ mole P bact$^{-1}$h$^{-1}$ | 0.45-26 | 0.11-32 | 0.13-11 | 0.17-23 | *0.1 - 28* |
| Per cell BP | $10^{-18}$g C bact$^{-1}$h$^{-1}$ | 46-136 | 25-60 | 3-17 | 1-14 | *4-23* |
| per BP LAP | nmol AA nmol C$^{-1}$ | 0.04-0.24 | 0.12-0.44 | 0.21-1.08 | 0.36-3.03 | *0.09-0.76* |
| per BP βGLU | nmol glucose nmol C$^{-1}$ | 0.003-0.017 | 0.007-0.034 | nd | nd | *nd* |
| per BP AP | nmol P nmol C$^{-1}$ | 0.09-2.3 | 0.05-11 | 0.46-8 | 0.6-40 | *0.04-1.7* |



Table 4. Turnover times of ectoenzymes (Km/Vm ratio). Means ± sd and range values given for all stations (n = 10). For leucine aminopeptidase (LAP), beta glucosidase (βGLU), alkaline phosphatase (AP). ld limits of detection, not enough data to plot Michaelis Menten kinetics. The turnover times are calculated from concentration kinetics using up to 50 µM concentration or up to 1 µM concentration.

|  | Units: days | surface | dcm | liw layers | mdw waters |
|---|---|---|---|---|---|
| Km50/Vm50 LAP | mean ± sd | 309 ± 214 | 144 ± 186 | 2912 ± 1756 | 4526 ± 2118 |
|  | range | 94-880 | 40-663 | 1294-6616 | 1308-7791 |
| Km1/Vm1 LAP | mean ± sd | 75 ± 28 | 41 ± 22 | 345 ± 235 | 634 ± 713 |
|  | range | 15-120 | 15-82 | 141-985 | 55-2481 |
| Km50/Vm50 βGLU | mean ± sd | 3464 ± 1576 | 3091 ± 1551 | ld | ld |
|  | range | 1997-7395 | 328-5481 | nd | nd |
| Km1/Vm1βGLU | mean ± sd | 126 ± 233 | 247 ± 273 | ld | ld |
|  | range | 20-784 | 15-873 | nd | nd |
| Km50/Vm50 AP | mean ± sd | 18 ± 20 | 39 ± 46 | 563 ± 542 | 1042 ± 1156 |
|  | range | 2-69 | 0.7 – 113 | 16 – 1441 | 20 – 3875 |
| Km1/Vm1 AP | mean ± sd | 5.6 ± 5.0 | 27 ± 37 | 268 ± 349 | 301 ± 172 |
|  | range | 1-17 | 0.6-106 | 12-1180 | 14-594 |



Fig 1

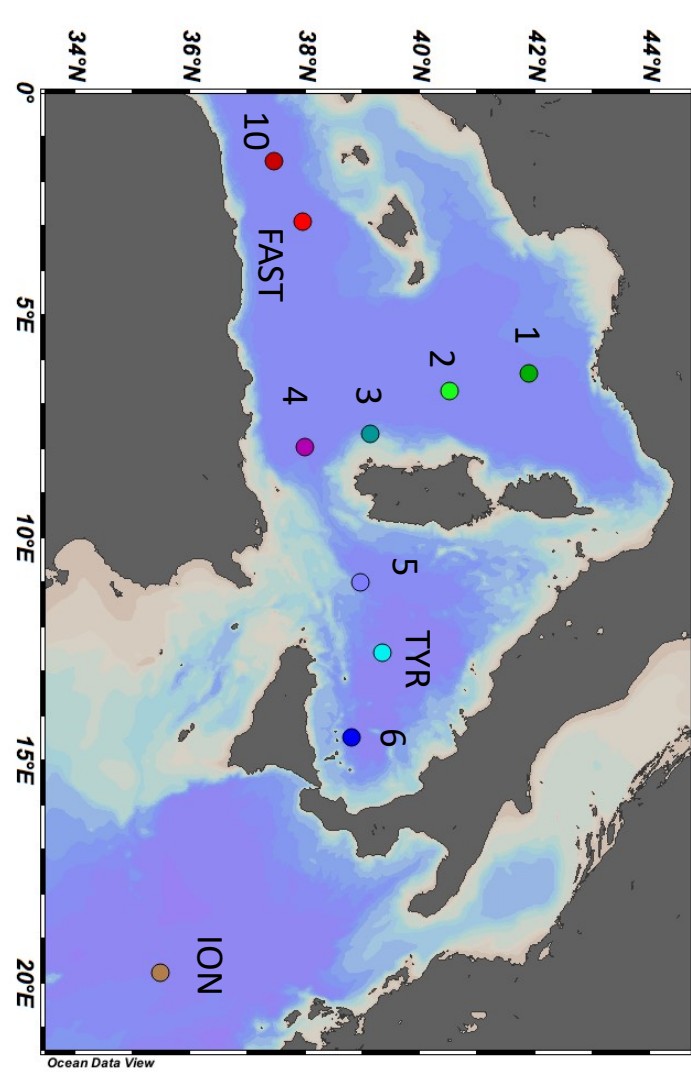



Fig 2

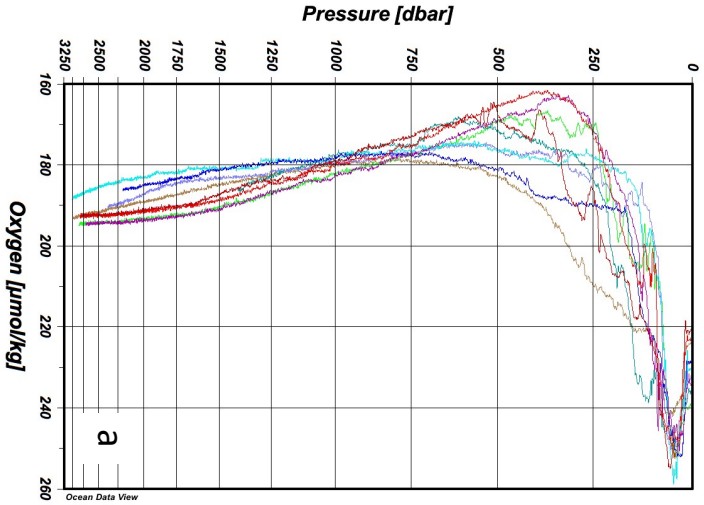

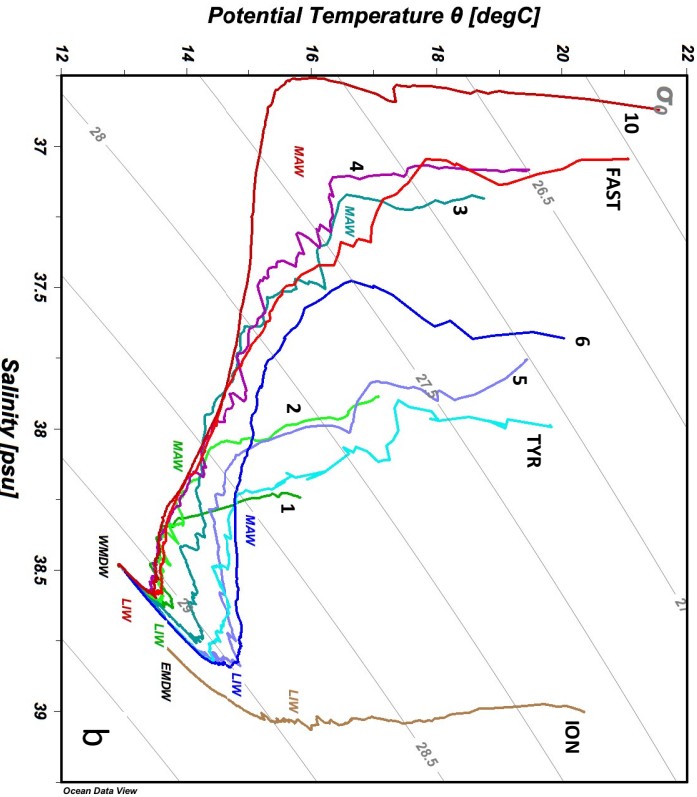



Fig 3

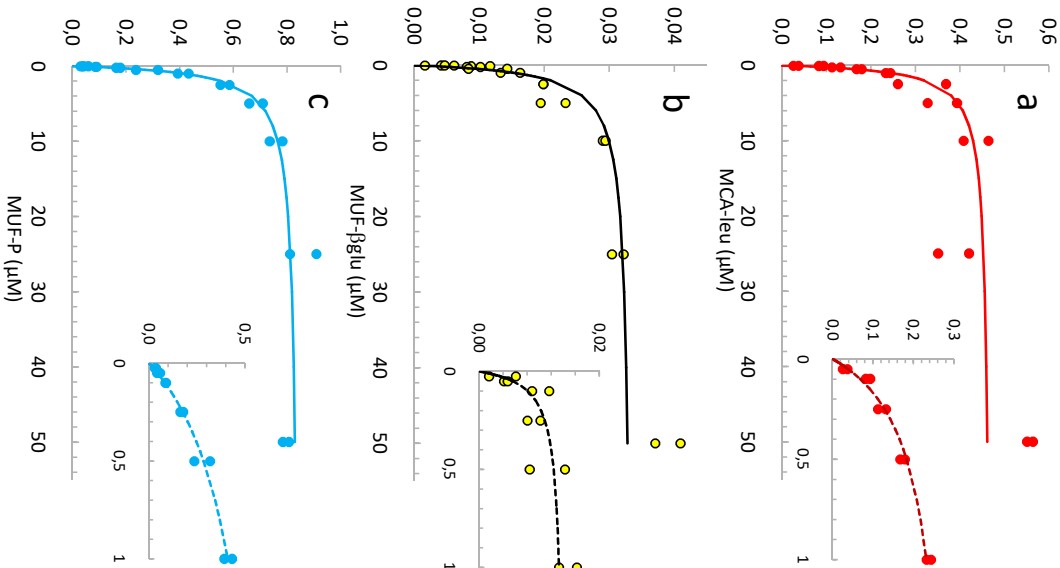





Fig 4

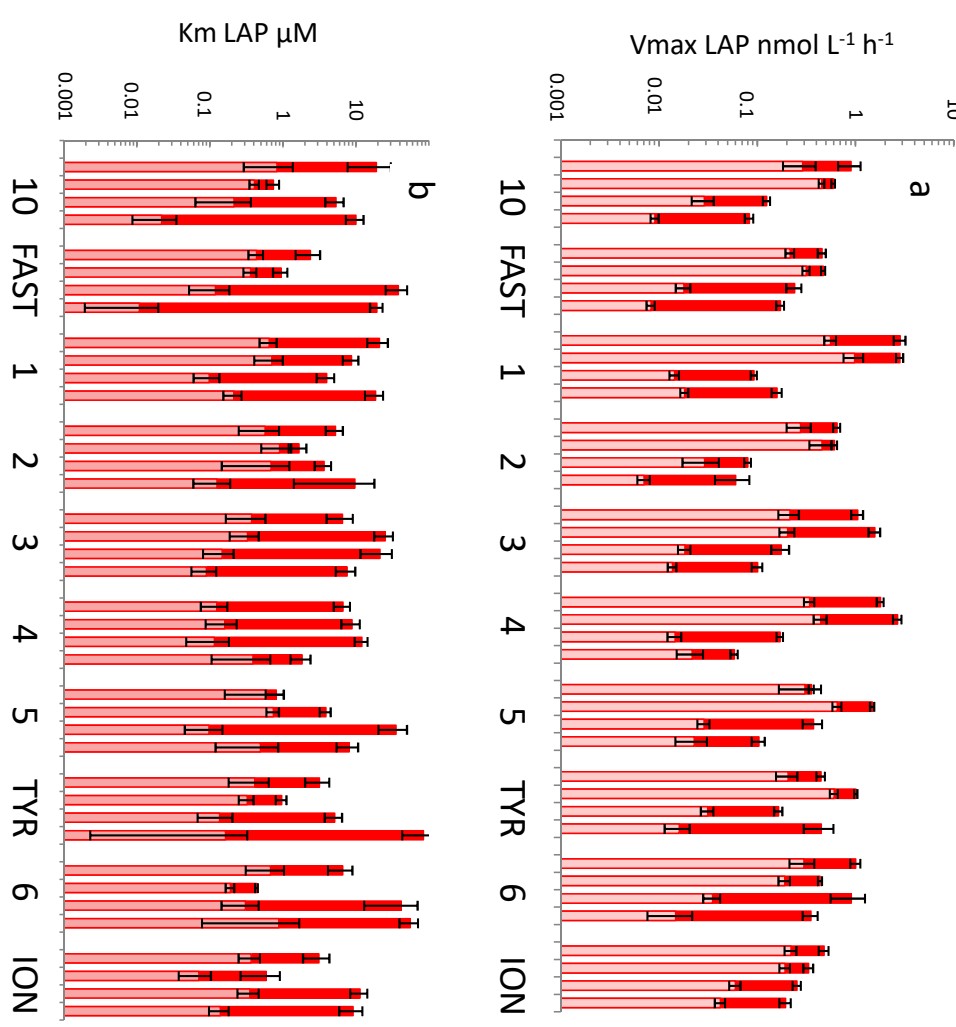

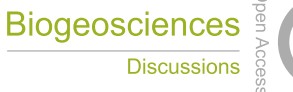

Fig 5

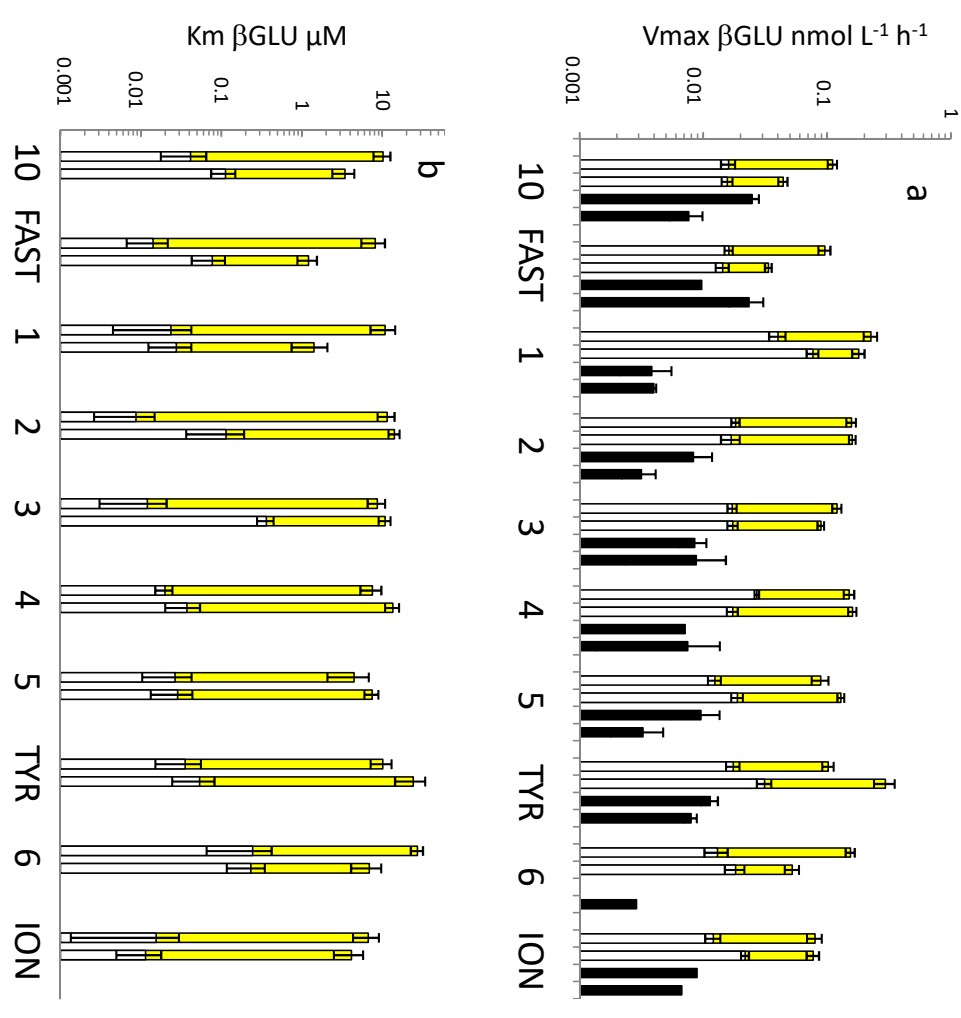



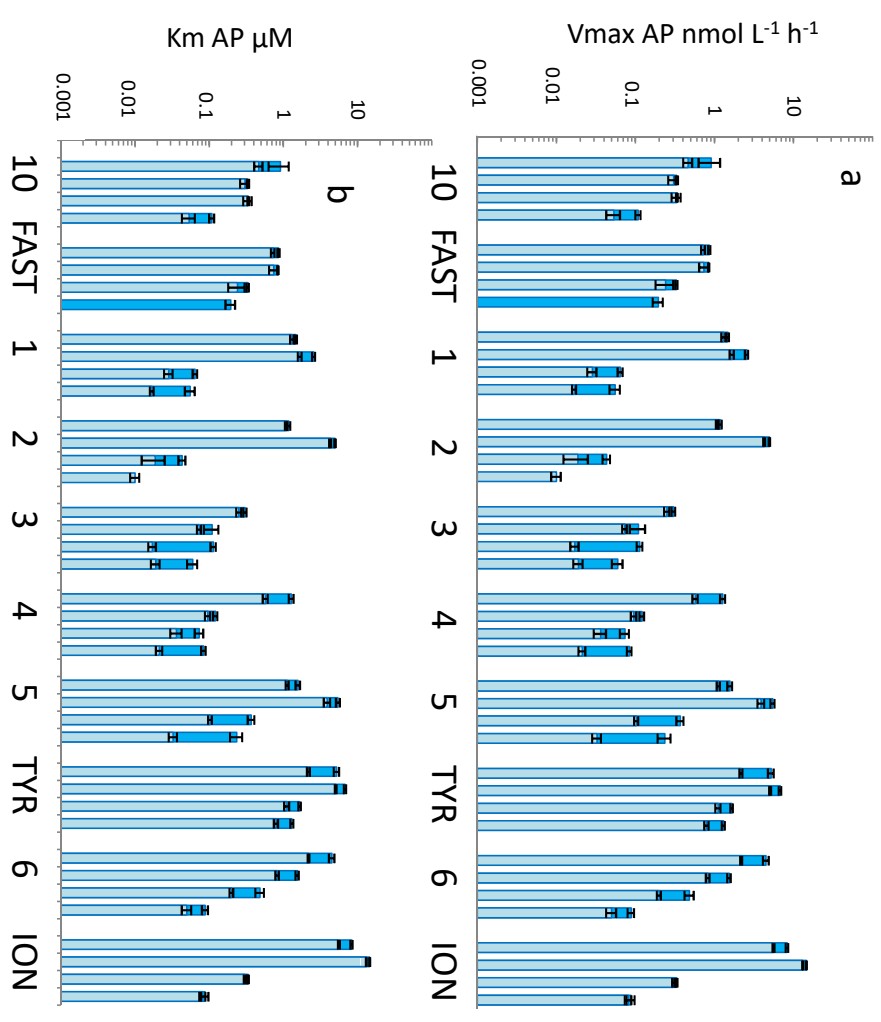



Fig 7

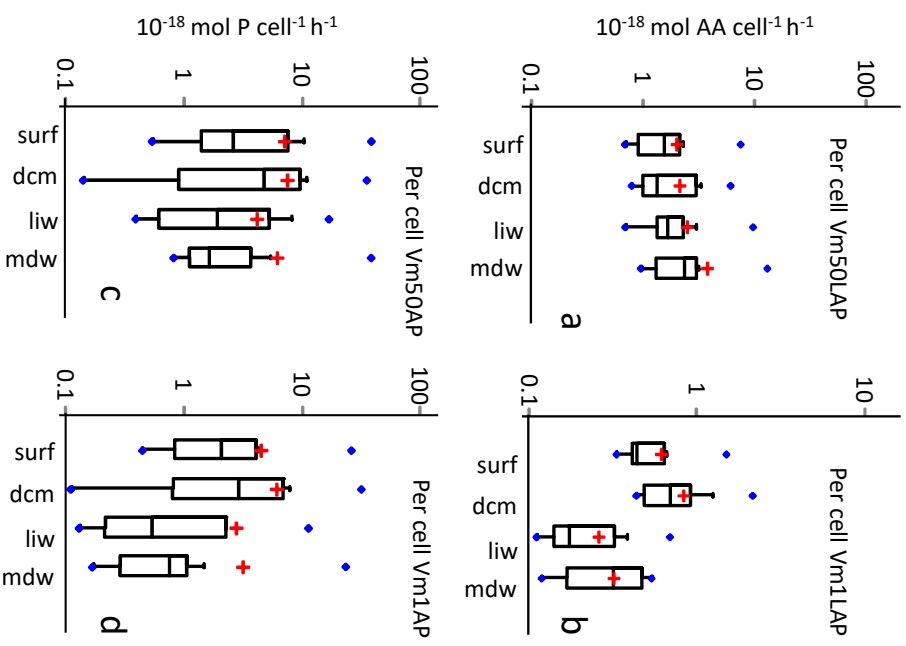





Fig 8

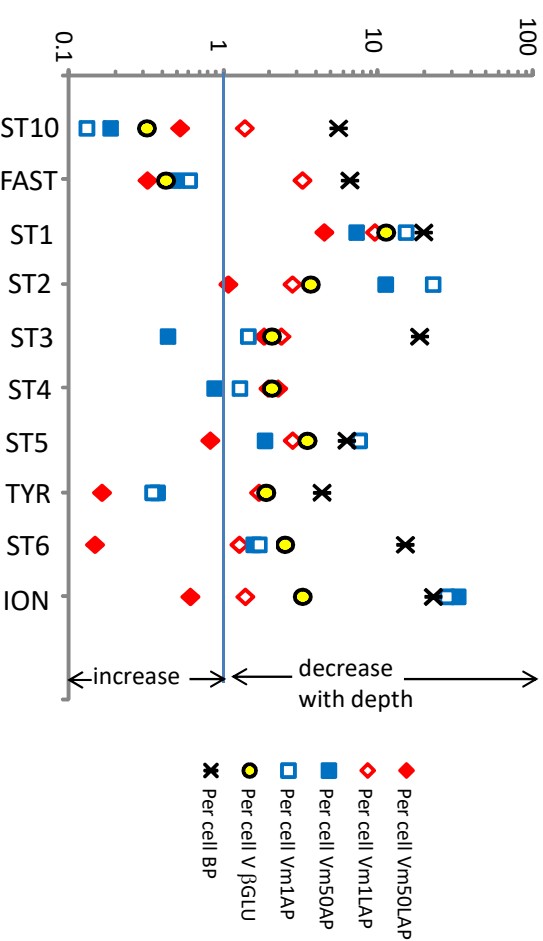



Fig 9



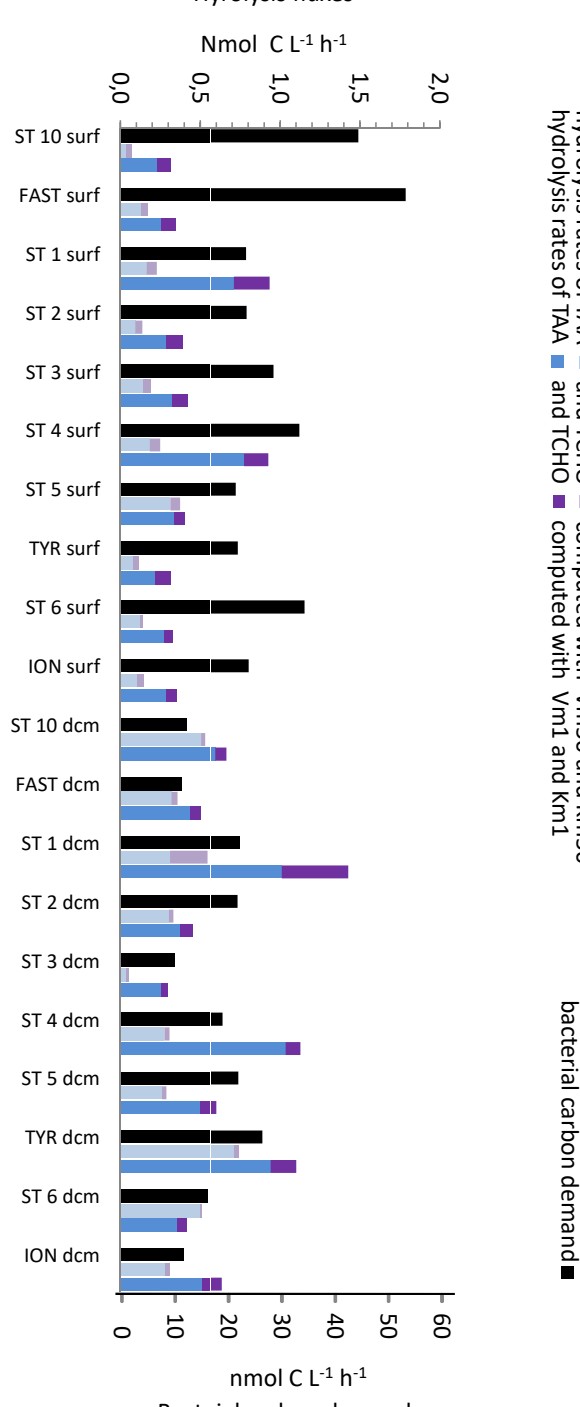

Fig 10