# Peer review of "Spatial patterns of ectoenzymatic kinetics in relation to biogeochemical properties in the Mediterranean Sea and the concentration of the fluorogenic substrate used."

_Biogeosciences, 2020_

## Referee Comment (RC1) · Anonymous Referee #1 · 7 Oct 2020

General comments: This article refers on the enzyme activity rates along the epipelagic and mesopelagic layers of a Mediterranean area, taking into consideration the variability of their kinetic properties in relation to the addition of different amounts of fluorogenic substrates. Particularly, two ranges are compared (0.025-1 $\mu$M with respect to 0.025-50 $\mu$M) to assess the degree of affinity between the enzyme and its substrate (in terms of Km Michaelis Menten constant). I think that the subject of this manuscript falls within the aims of Biogeosciences, This paper presents a large dataset of enzyme kinetics in different regions of the Mediterranean; their interpretations and conclusions are quite good, the supplied figures are explicative and the list of references is consistent, however the English language at some points is not very clear and could be

improved. Moreover, the structure of the Introduction could be improved as suggested in the Specific comments. However, the major doubt I have regarding its acceptance is that the Special Issue Atmospheric deposition in the low-nutrient-low-chlorophyll (LNLC) ocean- where it has been submitted - has a focus different from the Special Issue main theme and the subject of this manuscript (enzyme expression and its variability depending on the added substrate concentrations) refers on the effects of the atmospheric deposition or aerosol on marine processes in 4 lines of the discussion only (from 714 to 717). Therefore in my opinion this represent a too short and insufficient discussion to justify the inclusion of this manuscript in the Special Issue above reported.

Specific comments: line 20, the definition of the depth range of epi- and meso-pelagic layers should be indicated; line 23, and throughout the text, I suggest to refer to low and high affinity enzymes (instead of affinity systems); lines 28-29, please check this sentence, it remains without a conclusion (probably "although" should be deleted); lines 32 and successive: current and past interpretation of the three tested enzymes and their relative differences regarding the choice of added....; line 35. I suggest to change into: were different depending on activity estimates were derived from high or low concentrations of substrates were used; line 42, is a subject ; line 51, for the enzyme kinetics (instead of concentration kinetic), the minimum substrate concentration.

The Introduction should be re-organized, since on line 60 the authors talk about multiphasic kinetics, and they again return on the concept of biphasic kinetic on line 69 and also on lines 76-77, in a logical order it would be better to move lines 69 and 76 close to line 60 to discuss about multiphasic kinetics Again, sentence on lines 88-90 could be moved close to lines 80-83 (both talking about the enzyme assay); line 59, within a single species; line 81, prior incubation of the sample with the substrate; line 92, especially regarding the origin; line 97, interaction among different enzymes; line 100, since in the Abstract the acronyms of the enzymes are reported, please use them for aminopeptidase and phosphatase; line 104, the kinetics of three enzymes targeting;

line 118, in this manuscript (in full); line 130, epipelagic (0-250), what reference do the authors used to select these depth ranges? please include it; line 142, remove comma after rosette; lines 182, 186, 324 , NO3 (3 in small character) line 182,Within epipelagic, nutrient (DIP and NO3) depleted layers; line 182, LWCC in full (liquid waveguide capillary cells); line 202, Two replicates per each TCHO sample; line 214, acetonitrile as solvent B was used; line 250, The amounts of the products MCA and MUF, released by LAP.....; line 255, were produced instead of dispatched; line 260, The hydrolysis rate was calculated; line 279, Vm1 and Km1; line 290 correlations among variables; line 290, Results are reported as means $\pm$ standard errors; line 296, remove (the) , changing seasonal pattern; line 308, 38.48) were different from the EMDW water mass being less salty and colder; line 310, please include a reference for the identification of water masses (according to their physical-chemical properties); line 327, The depth of dcm (use acronym); line 333, the mean values of DOC/DON, line 334, the mean values of TAA; line 337, decreased from epipelagic to deeper waters; line 338, contribution of TAA-N to DON which ranged from 3 to 9% at surf and dcm and from 1.6 to 4.6% at liw and mdw respectively; line 342, according to; line 344, varied among the stations; line 345, the highest values were measured; line 352, were determined over highly variable trophic conditions; line 357, the finding that measurements at liw and mdw layers for GLU were below detection limits; line 365, please indicate how the CV coefficient of variation was calculated(mean/st. dev x100); line 370 AP Vm50 and Vm1 mean values; line 382, variable patterns (instead of inconsistent); line 390, their differences between two sets of concentrations (Vm50 and Vm1) were; line 394, with saturation rates occurring around 1 $\mu$M of added MUF-P (this repeats line 355); line 403, about the declining trend; line 411, Km 50/Km1 ratio (Fig.6b) decreased with depth, so enzyme affinity for the substrate increased (please check this sentence); line 417, the turnover times were the shortest and the longest; line 439, the mean values (instead of medians); line 440 and in Tab. 3, gC cell (instead of bact) ; Tab. 2 caption: Microbial abundances, activity rates and kinetics measured at the 4 layers first row, surface, dcm. liw. mdw (remove layers and waters); Tab.3 caption: in deep waters (liw and mdw , per BP/LAP

in nmolAA/nmolN (not C) and per BP/AP in nmolP/nmol P (not C); Tab. 4, first row: surface, dcm, liw, mdw (remove layers and waters); Fig. 8, per cell V50 or V1 GLU?; Fig. S1, include labels for a and b; lines 442-443,please check this sentence: while the specific activities per unit cell decreased with depth, the activity per unit BP increased (but from 46-136 they change into 1-14), since in Figure 8 a decrease with depth instead of an increase is shown ; line 444, according to stations; line 449, increased with depth (include reference to Fig.8) ; A better title for Paragraph 3.5 should be Hydrolysis versus carbon/nitrogen demand; line 452 higher than TAA concentrations (Table 2, Table S1); line 485, the differences between these two LAP systems could reach the differences?.(explain better this statement); lines 478 and 489, the difference between the two concentration sets of substrates (not two types of enzymes); It sounds very strange that the <0.2 micron fraction exceeded the total enzyme activity! how do the authors explain this strange result? line 508, expression of enzyme activities instead of development; lines 507-509, re-write or delete this sentence, it is unclear; line 529, This increases the importance of Km1 at low concentrations of substrate (is it right the meaning? just to avoid to repeat the risk of overestimation, already written in line 530); line 530, difference in the response; please check the references: Siokou-Frangou et al. 2010 (not included in the reference list), Zaccone et al. (2010) not cited in the text; line 555, remove for these authors; line 560, These authors suggested; line 604 (Limit of detection, not acronym); line 661 LLBN? please report in full; rewrite sentence on line 675; the direct influence on the determination of Km and Vm of the use of an appropriate set...; lines 679,680, 682, 683, 694 instead of affinity system I suggest affinity enzyme; line 691 Lemée 2002 or 2012?; line 704, supported by peptides and polysaccharides hydrolysed by enzyme activities; line 705, please report BGE in full (no acronym has been reported before)

---

## Referee Comment (RC2) · Anonymous Referee #2 · 26 Oct 2020

The paper "Spatial patterns of biphasic ectoenzymatic kinetics related to biogeochemical properties in the Mediterranean Sea." by France Van Wambeke et al., reports prokaryotic ectoenzymatic activity, abundance and heterotrophic production in the epipelagic and the upper part of the mesopelagic layers in the Mediterranean Sea. In this study, the Vm and Km of the 3 enzymes (alkaline phosphatase (AP), aminopeptidase (LAP) and $\beta$-glucosidase ($\beta$GLU)) were determined using 2 series of substrate concentration. The paper points out that the choice of substrate concentrations affect the results and their interpretation.

Strength points

The paper presents an impressive quantity of data on ectoenzymatic activity of 3 enzymes: alkaline phosphatase (AP), aminopeptidase (LAP) and $\beta$-glucosidase ($\beta$GLU). The data are of good quality and I think they deserve publication. The methods are detailed and well described. This paper has the potential to be a reference work for future studies on enzymatic activity, since the authors showed that the use of different ranges of concentrations of substrates give different results and therefore lead to bias in the interpretation of the results. I think this is a very important point, since the measure of enzymatic activity is crucial to get insights into the main biogeochemical fluxes in the oceans. Often the published data are difficult to compare due to the different concentrations of substrate used leading to contrasting interpretations.

Weak points

The main shortcoming is the language, often due to language issues, I did not understand what the authors means and the paper is confusing in many points. There are also some grammatical mistakes and typos. I think that there are too many details in the results making them hard to follow. The discussion is not focused, there are too much information that confound the reader. My main recommendation is to deeply revise the English, to simplify the results and to rework the discussion, in order to make the paper easier to follow and to highlight its main message.

Specific comments are reported below.

Introduction

All the introduction would benefit of a deeply revision of the English. The text does not flow well and there are some grammatical issues making hard to read it.

L38-40, P1. "Most of the organic matter being in the state of high molecular weight material, its hydrolysis by ectoenzymes plays an important role in the degradation, utilization and mineralization processes in aquatic environments, but also in nutrients regeneration (Hoppe, 1983; Chróst, 1991)." I think there is a grammatical issue in this

sentence.

L47 Seldomly?

L49-51, P2. "Within these 5 studies for the concentration kinetic the minimum 50 concentration used was 50 nM at the lowest, ..." I think there is a grammatical issue in this sentence.

L66, P2. "among 44 isolated strains", strains of what? Bacteria?

L95-97, P3. "The interaction between different enzymes has been largely studied in the Mediterranean Sea (Zaccone and Caruso, 2019) due to the particular role of this elemental stoichiometry." I do not understand this sentence. What do you mean with "interaction among enzymes"? How can it affect the C/N/P ratios of nutrients and organic matter?

L107-109, P3. "Our aim was to study the effects of the respective activities of the ectoenzymes in relation to the quality of the organic matter present, below the productive layer and above the deep Mediterranean waters" How was the quality of the organic matter investigated? I did not find anything about it in the paper.

L104-107, P3. "In this study, we investigated in the Mediterranean Sea, the kinetics of three series of enzymes targeting proteins, phospho-mono esters and carbohydrates (aminopeptidase, alkaline phosphatase 105 and $\beta$-D -glucosidase respectively) in relation to the elemental stoichiometry of particulate and dissolved organic matter." There is only 1 line (L333, P8) about stoichiometry data in the results and few lines (L572-573 and 592-593) in the discussion, so I think that this cannot be one of the main goals of the paper.

L106-107, P3. "We have paid particular attention to the use of a wide range of substrates concentrations to evaluate potential multiphasic kinetics." and L116-117, P 3 "Finally, we discuss the biases in interpretation of past and current enzymatic kinetic, potentially induced by the reduced range of used substrates concentration." I think

that the use of different concentration ranges of substrate for the 3 enzymes leading to different results is the main message and I would focus the manuscript on it.

2. Materials and Methods

L143, P4. "Heterotrophic prokaryotic production (BP), heterotrophic prokaryotic abundances (BA)" I would use HPA and HPB as abbreviations.

L147, P4. Replace ectoenzymatic activities by EEA, since the abbreviation is defined at L143-144.

L153, 154, 156 and 157, P4. I would use upper case letters for the abbreviations, in particular for DCM, LIW and MDW.

L156-157, P4. "and second sampled at 1000 m (the limit between meso and bathy-pelagic waters), except at 2 stations (FAST, 2500 m; ION, 3000 m) named 'mdw' " Why did you select 1000 m as depth representative of deep waters and not a sample collected close to the bottom? How do you think that the different sampling depth at station FAST and ION can affect the results? This should be discussed in section 4.2 and 4.3.

L192-193, P5. Please add the batch of the CRM you used for DOC analysis, its nominal and measured concentration.

L217, P5. "Bacterial production (BP, sensus stricto referring to prokaryotic heterotrophic production)", BP was already defined at L143, P4.

L225, P6. "On 9 occasions", do you mean replicates? Samples?

Results

The results are very heavy to read, in section 3.3 there are too many comparisons, too many details that are not relevant for the main message of the paper. I recommend to simplify this section and to avoid details not relevant for the main message of the paper.

3.1 Hydrological situations.

The title should be changed, what does "Hydrological situations" mean? I suggest Physical properties

This section should cite some papers reporting the circulation of the Med Sea and the main physical properties of the water masses.

L294-295, P7. "The sampled stations have basins and latitude characteristics that were superimposed on a changing the seasonal pattern", I do not understand this sentence.

L299, P7. "Modified Atlantic Waters (MAW) are characterized by low salinity below the seasonal thermocline" Do you mean above the seasonal thermocline?

L299-301, P7. "this property is stretched in the westernmost stations, then progressively relaxes on eastern station, revealing an eastward circulation in the Algerian Basin and a dispersion in the connected basins (northwestern Mediterranean, Tyrrhenian, and Ionian Seas)." And L303-305, P7. "This property is pronounced in the eastern stations and progressively lowered on the western stations, revealing an opposite circulation pattern to the MAW. " It is not possible to infer the water mass circulation from the T/S graphs. Please add references and rework the sentences.

L305-308, P7. Please add references.

L308-310, P8. Please add references.

L310-311, P8. "The core of LIW is characterized by lower oxygen content than its surrounding water masses, shallower (MAW) and deeper (WMDW and EMDW)". Looking at figure 2a, this observation is not always true, for example at stations ION and 6, LIW has higher oxygen than deep waters.

L312-314 "We thus presented all the figures/tables in the order ST10, FAST, ST1, ST2, ST3, ST4, ST5, TYR, ST6 and ION, according to the expected circulation of the LIW (from the right to the left)." If you want to follow the LIW path, I think it is better to invert

the order of the stations, since LIW move from ION to St.10.

3.2 Biogeochemical situation. I would replace situation with properties.

In this section, the values of DOC, DON and DOP should be reported not only their change with depth.

Section 3.3. Ectoenzymatic activities – kinetic trends

This section is really heavy and in some parts there is not correspondence between the number in the text and in the tables. The authors should carefully rework this section deleting all the details that are not relevant and checking the correspondence between the number in the text and in the tables.

L352, P8. "The ectoenzymatic activities were determined using large trophic conditions", I do not understand this sentence.

L368-L374, P9. Is this paragraph relevant? Looking at table 2, I find different numbers.

L376-380, P9. "For LAP (Fig. 4), Vm50 was on average 3 times higher than Vm1 in 'surf' and 'dcm' layers, but the differences between these two rates increased with depth (x9 in 'liw', x12 in 'mdw'). Vm50 decreased from epipelagic to mesopelagic waters by a factor of 8 on average, (ratio 'depth variation factor' – DVF), but by a factor x19 for Vm1 (Fig. 4a)." It is very hard to see these differences in Figure 4. I think you should also refer to Table 2. Looking at table 2 the number are different. As an example: Vm50 was 10-times not 12 higher that Vm1 in mdw. "Vm50 decreased from epipelagic to mesopelagic waters by a factor of 8 on average,", if I consider dcm epipelagic and LIW mesopelagic Vm50 decreases by 4.6 times, if I consider surf as epipelagic the decrease is of 3.8, so I don't understand how the authors calculated a value of 8, the same for Vm1.

L383, P9. I think there is a typo SD10, SD2 and SD6 should be St10, St.2 and St.6. St. FAST and St.1 are missing, they also show Km50 of LAP lower at the dcm than in the surf.

L395-396, P9. Also here there is not correspondence among the numbers in the text and the numbers that I can calculate from the table. Please check.

L410-411, P10. "Average Km50/Km1 ratio for $\beta$GLU was 320 and average Km1/Km50 ratio for LAP was 118." My calculation looking at table 4 indicate 240 instead of 320 and 79 instead of 118. Please check.

L436, P10. I think you should cite Fig.7c,d not Fig.7a,b.

Discussion

The discussion is confounding, there are a lot of interesting ideas, but they are lost in the text. The discussion would strongly benefit of deeply revision of the English. I think that section 4.1 should be the focus of the paper, but I miss a conclusion. From your data do you suggest to use Vm1 or Vm50? Km1 or Km50? Or since they give different information does the use of one or the other depend on the goal of the work? From my understanding the use of a not-appropriate range of substrate determine bias in the results and in the interpretation and data, obtained using different range of substrate, are not comparable. I think these points should be better stressed in this section. I also think that the other sections should support this one, showing how the different ranges may affect the interpretation of trends with depth and regional variability.

L481, P11. "The biphastic factor as defined in Tholosan et al (1999). "Please define it in the text to help the reader.

L505-509, P12. Please rework, I do not understand this sentence.

L535-539, P13. This paragraph should be moved in section 4.1.

L569-571, P13. "With concentration kinetics ending at 50 $\mu$M of MUF-P, the specific activities of AP reached using per cell Vm50 or per cell Vm1 were not so different and their trend with depth were similar (Fig. 8). " It is really hard to see these trends in Fig. 8.

L572-573, P14. "whereas DOC/DOP ratio decreased (from 2200-2400 to 1500-1200), suggesting a preference for heterotrophic prokaryotes to use dissolved organic phosphorus as substrate of AP." Usually DOC/DOP ratio increases with depth due to the preferential removal of P. Are these ratios calculated using data collected in this cruise? The removal of DOP by heterotrophic prokaryotes should increase the DOC/DOP so, this sentence has no sense to me.

L574-587, P14. This paragraph should be moved in section 4.1.

L595-611, P14. This paragraph is not clear to me.

4.3 Regional variability

L613-615, P14. "In epipelagic waters, both AP maximum rates (Vm1, Vm50) significantly increased from the Algerian/Ligurian Basins to the Tyrrhenian Basin (t test, p = 0.002 and p = 0.02, respectively), and reached maximum values at ION.". In Figure 6 this pattern is not very clear.

L658-663, P15-16. This paragraph should be moved in section 4.1.

L675-687, P16. This paragraph should be moved in section 4.1.

Figure 2. Please enlarge the name of water masses. I think the value of this figure would strongly increase if you mark on the T/S graphs where the samples for enzymatic activity were collected. You could also add a zoom of intermediate and deep waters.

Figure 4-6. Please add the legend.

Figure 8. This figure is really confuse and hard to understand, I suggest to remove it.

Figure 9 and 10. I suggest to report in the caption how bacterial nitrogen demand and bacterial carbon demand was calculated.

Table1, Please check the title of the columns.

Table 2, line 6 column 5, I think the lines are inverted, if not please check the number

in the text.

---

## Author Comment (AC1) · 20 Nov 2020

**Response to referee 1 "Spatial patterns of biphasic ectoenzymatic kinetics related to biogeochemical properties in the Mediterranean Sea" by France Van Wambeke et al. ms BG-2020-253**

**Anonymous Referee #1 and our responses in blue**
**General comments:**
This article refers on the enzyme activity rates along the epipelagic and mesopelagic layers of a Mediterranean area, taking into consideration the variability of their kinetic properties in relation to the addition of different amounts of fluorogenic substrates. Particularly, two ranges are compared (0.025-1 μM with respect to 0.025-50 μM) to assess the degree of affinity between the enzyme and its substrate (in terms of Km Michaelis-Menten constant). I think that the subject of this manuscript falls within the aims of Biogeosciences, This paper presents a large dataset of enzyme kinetics in different regions of the Mediterranean; their interpretations and conclusions are quite good, the supplied figures are explicative and the list of references is consistent,

We are grateful to the referee 1 for reviewing the manuscript. We appreciate all his/her insightful comments and we acknowledge that they have encouraged us to simplify and improve the manuscript. Please find below a detailed answer to all the raised questions and comments

However the English language at some points is not very clear and could be improved.
The revised ms has been proofread by a native English speaker

Moreover, the structure of the Introduction could be improved as suggested in the Specific comments.
We reorganized the introduction, find below our reply to specific comments

However, the major doubt I have regarding its acceptance is that the Special Issue Atmospheric deposition in the low-nutrient-low-chlorophyll (LNLC) ocean- where it has been submitted - has a focus different from the Special Issue main theme and the subject of this manuscript (enzyme expression and its variability depending on the added substrate concentrations) refers on the effects of the atmospheric deposition or aerosol on marine processes in 4 lines of the discussion only (from 714 to 717). Therefore in my opinion this represent a too short and insufficient discussion to justify the inclusion of this manuscript in the Special Issue above reported.

The main reason why this manuscript has been submitted to the above cited Special Issue is that the presented data were obtained during the cruise PEACETIME. As stated in the description of the SI: "(…) It will present the results obtained during the PEACETIME (ProcEss studies at the Air-sEa Interface after dust deposition in the MEditerranean sea) cruise conducted in 2017 in the Mediterranean Sea, but the special issue is open to any submissions provided that the subject is consistent with the objectives defined above."
 The other reason is that ectoenzymatic activities presented here are also the base of the discussion of 2 other papers fully related with the topic of the special issue: one has recently been submitted (Van Wambeke et al., bg-2020-411) and the other one is in preparation (Pulido-Villena et al., in prep). However, we shall leave this decision to the Editors of the special issue.

Van Wambeke, F., Taillandier V., Desboeufs, K., Pulido-Villena, E., Dinasquet, J., Engel, A., Marañón, E., Ridame, C., Guieu, C.: Influence of atmospheric deposition on biogeochemical cycles in an oligotrophic ocean system, Biogeosciences Discuss., https://doi.org/10.5194/bg-2020-411, in review, 2020.

Pulido-Villena, E., Van Wambeke, F., Desboeufs, K., Petrenko, A., Barrillon, S., Djaoudi, K., Doglioli, A., D'Ortenzio, F., Fu, Y., Gaillard, T., Guasco, S., Nunige, S., Raimbault, P., Taillandier, V., Triquet, S., Guieu, C. Phosphorus cycling in the upper waters of the Mediterranean Sea (Peacetime cruise): relative contribution of external and internal sources, in prep for Biogeosciences, special issue PEACETIME.

**Specific comments:**

line 20, the definition of the depth range of epi- and meso-pelagic layers should be indicated
Instead, we have specified the 4 depths sampled as follows:
'Ectoenzymatic activity, prokaryotic heterotrophic abundances and production were determined in the Mediterranean Sea. Sampling was carried out in the sub surface, the deep chlorophyll maximum layer, the core of the Levantine Intermediate waters and the deeper part of the mesopelagic layers.'

line 23, and throughout the text, I suggest to refer to low and high affinity enzymes (instead of affinity systems)
This is done

lines 28-29, please check this sentence, it remains without a conclusion (probably "although" should be deleted).
The sentence was modified as:
'The contribution of ectoenzymatic hydrolysis to the heterotrophic bacteria requirements was high in terms of N, but it was low in terms of C.'

lines 32 and successive: current and past interpretation of the three tested enzymes and their relative differences regarding the choice of added....;
We have rephrased the sentence to make it clearer:
'This study clearly highlights the bias in current and past interpretation for the kinetic parameters of the 3 enzymes according their fluorogenic substrate concentration sets.'

line 35. I suggest to change into: were different depending on activity estimates were derived from high or low concentrations of substrates were used;
The sentence was modified as:
'In particular, aminopeptidase/βglucosidase ratios, and some depth trends, were different depending on whether activities estimates were derived from high or low concentrations of the fluorogenic substrates.'

line 42, is a subject This is done

line 51, for the enzyme kinetics (instead of concentration kinetic), the minimum substrate concentration. This is done

The Introduction should be re-organized, since on line 60 the authors talk about multiphasic kinetics, and they again return on the concept of biphasic kinetic on line 69 and also on lines 76-77, in a logical order it would be better to move lines 69 and 76 close

to line 60 to discuss about multiphasic kinetics Again, sentence on lines 88-90 could be moved close to lines 80-83 (both talking about the enzyme assay);
This part of the introduction has been re-organized following the referee comments.

line 59, within a single species This is done

line 81, prior incubation of the sample with the substrate This is done

line 92, especially regarding the origin This is done

line 97, interaction among different enzymes This is done

line 100, since in the Abstract the acronyms of the enzymes are reported, please use them for aminopeptidase and phosphatase;
In the revised version, there are not anymore abbreviations neither in the abstract nor in the introduction and they are defined here in Material and methods section 2.1.

line 104, the kinetics of three enzymes targeting This is done

line 118, in this manuscript (in full) This is done

line 130, epipelagic (0-250), what reference do the authors used to select these depth ranges? please include it;
The sentence was modified as:
'Generally, at least 3 casts were conducted at each short station. One focused on the first 250 meters and the second one on the whole water column….. The third cast…'

 line 142, remove comma after rosette This is done

 lines 182, 186, 324 , NO3 (3 in small character)
We assumed this is just a matter of the choice of the abbreviation term. To be formerly correct we should use nitrate, or $NO_3^-$. For simplicity to avoid abundant superscripts and subscripts in the ms, we decided to abbreviate nitrate as NO3. To be clear for the reader we modified the sentence in sub-section 2.2 as:
'Nitrate (abbreviated as NO3),…'

line 182,Within epipelagic, nutrient (DIP and NO3) depleted layers;
The sentence was modified as:
'Within the epipelagic surface layers, DIP and NO3 were determined using the liquid waveguide capillary cell method (LWCC)…'

line 182, LWCC in full (liquid waveguide capillary cells); This is done

line 202, Two replicates per each TCHO sample; This is done

 line 214, acetonitrile as solvent B was used; This is done

line 250, The amounts of the products MCA and MUF, released by LAP.....; This is done

line 255, were produced instead of dispatched; This is done

line 260, The hydrolysis rate was calculated; This is done

line 279, Vm1 and Km1; This is done

line 290, Results are reported as means ± standard errors; This is done

line 290 correlations among variables; This is done

line 296, remove (the) , changing seasonal pattern; This is done

line 308) were different from the EMDW water mass being less salty and colder; This is done

[revised manuscript text omitted]

The Mermex Group. Marine ecosystems' responses to climatic and anthropogenic forcings in the Mediterranean. Progress in Oceanography, 91(2), 97–166, doi: 10.1016/j.pocean.2011.02.003, 2011

Wust, G.: On the vertical circulation of the Mediterranean Sea. Journal of Geophysical Research, 66, 10, 3261-3271, 1961

line 327, The depth of dcm (use acronym); This is done

line 333, the mean values of DOC/DON, This is done

line 334, the mean values of TAA; This is done

line 337, decreased from epipelagic to deeper waters; This is done

line 338, contribution of TAA-N to DON which ranged from 3 to 9% at surf and dcm and from 1.6 to 4.6% at liw and mdw respectively;
This sentence was modified as:
'…this trend was confirmed by the ratio of TAA-N to DON (Fig. S1a) which decreased significantly with depth (p < 0.001).'

line 342, according to; This is done

line 344, varied among the stations; This is done

line345, the highest values were measured; This is done

line 352, were determined over highly variable trophic conditions; This is done

line 357, the finding that measurements at liw and mdw layers for GLU were below detection limits
The sentence was modified as:
'For 'liw' and 'mdw' layers computation of βGLU kinetics was impossible as only a few time series gave a significant linear increase of fluorescence with time, when adding 25 or 50 µM fluorogenic substrate.'

line 365, please indicate how the CV coefficient of variation was calculated (mean/st. dev x100);
The definition of CV was added in the Material and Methods section 2.5 'statistics'

line 370 AP Vm50 and Vm1 mean values; This is done

line 382, variable patterns (instead of inconsistent); This is done

line 390, their differences between two sets of concentrations (Vm50 and Vm1) were; This is done

line 394, with saturation rates occurring around 1 µM of added MUF-P (this repeats line 355);
The first sentence of the paragraph describing AP was modified as:
'AP was the enzyme for which Vm1 and Vm50 were the closest (averages of Vm50/Vm1 ratio for the whole data set was $1.6 \pm 0.5$) (Fig. 6a), showing that saturation rates occurred at 1 µM of added MUF-P (Fig. 3)'.
This was not repeated anymore in this paragraph.

line 403, about the declining trend; This is done

line 411, Km 50/Km1 ratio (Fig.6b) decreased with depth, so enzyme affinity for the substrate increased (please check this sentence);
Line 411 The sentence written is 'It was observed that the general trend was that Km50 increased more with depth (DVF > 0 at 7 stations, ranging from x2 to x29) than Km1 (DVF > 0 at 9 stations, ranging x1.9 to x3.8, see ST1 and ST5'
These statements are correct, but there was a mistake in Fig6b. Indeed we plotted 2 times the Fig 6a (VmAP) instead of Fig 6a (VmAP) and 6b (KmAP). We apologize for this mistake, the correct Figure 6b is presented below and in accordance with what was described line 411.

[Figure]

line 417, the turnover times were the shortest and the longest; This is done

line 439, the mean values (instead of medians);
Figure 7, to which line 439 refers to, represents Tukey box plots and, thus, the middle point is actually the median value of the dataset. This is why I talk about medians.

line 440 and in Tab. 3, gC cell (instead of bact) ; This is done

Tab. 2 caption: Microbial abundances, activity rates and kinetics measured at the 4 layers
The sentence was modified as:
'Heterotrophic bacterial abundances (BA), bacterial production (BP) and ectoenzyme kinetics for leucine aminopeptidase (LAP), β-glucosidase (βGLU) and alkaline phosphatase (AP) at the 4 layers studied for ectoenzymatic activities.'

first row, surface, dcm. liw. Mdw (remove layers and waters); This is done

Tab.3 caption: in deep waters (liw and mdw , This is done
per BP/LAP in nmolAA/nmolN (not C) and per BP/AP in nmolP/nmol P (not C);
The last 3 specific rates of this Table 3 are Vm of ectoenzymatic activities calculated per unit BP, BP expressed in carbon units.

Tab. 4, first row: surface, dcm, liw, mdw (remove layers and waters); This is done

Fig. 8, per cell V50 or V1 GLU?;
It is not Vm50 nor Vm1 as we could not make kinetics. This is why it is just written 'V'. This is explained in the legend:
'For β−glucosidase DVF, specific activities are based on the few detectable rates at high concentration (per cell Vβglu, yellow dots).'
For clarity we also separated Fig 8 in three sub-plots, see response to referee 2.

Fig. S1, include labels for a and b; This is done, as well as for Fig. S2

lines 442-443,please check this sentence: while the specific activities per unit cell decreased with depth, the activity per unit BP increased (but from 46-136 they change into 1-14), since in Figure 8 a decrease with depth instead of an increase is shown ;
-       LAP per cell based on Vm1 decreased with depth (DVF 1.3-9.6, Table 3, Fig. 8),
-       LAP per cell based on Vm50 increased with depth (Fig. 8)
-       LAP per unit BP based on Vm1 increased with depth (DVF 0.09-0.76, Table 3)
The second sentence was modified as:
'While the specific LAPVm1 per unit cell decreased with depth, the specific LAPVm1 per unit BP increased with depth at all stations (Table 3, Fig. 8a).'

line 444, according to stations; This is done

line 449, increased with depth (include reference to Fig.8) ; This is done

A better title for Paragraph 3.5 should be Hydrolysis versus carbon/nitrogen demand;
We left the title unchanged. Indeed, in this sub-section, we estimate ectoenzymatic *in situ* hydrolysis rates (defined and calculated as in Material and Method section, last paragraph in sub section 2.4) but we do not compare them to carbon or nitrogen demand. This is done later on, in the discussion section.

 line 452 higher than TAA concentrations (Table 2, Table S1); This is done

line 485, the differences between these two LAP systems could reach the differences?.(explain better this statement);
The sentence was modified as: 'The two LAP enzymatic systems observed in the water column could reach a difference as large as that found in the sediment (Tholosan et al., 1999), where large gradients of organic matter are found

lines 478 and 489, the difference between the two concentration sets of substrates (not two types of enzymes);
This sentence was removed

It sounds very strange that the <0.2 micron fraction exceeded the total enzyme activity! how do the authors explain this strange result?
It is probably an artifact of the filtration. A minimum of vacuum pressure is necessary and some cells can be broken during filtration, thus release of intracellular enzyme may occur. Anyway, as we simplified the results section by removing details, this part was reduced as follows:
'During the PEACETIME cruise we ran some size fractionation experiments in 'surf' and 'dcm' layers (results not shown). The contributions of the < 0.2 μm fraction to the bulk activity was on average 60 ± 34 % (n = 12) for AP, 25 ± 16 % (n = 12) for βGLU and 41 ± 16 % (n = 12) for LAP, confirming these trends in the Mediterranean Sea'

line 508, expression of enzyme activities instead of development; lines 507-509, re-write or delete this sentence, it is unclear;
The sentence was removed and this paragraph started by:
'LAP enzymatic systems showed, in opposition to AP, more differences and different trends with depth.'

line 529, This increases the importance of Km1 at low concentrations of substrate (is it right the meaning? just to avoid to repeat the risk of overestimation, already written in line 530); We agree, there is repetition, we removed the first sentence

line 530, difference in the response; This is done

please check the references: Siokou-Frangou et al. 2010 (not included in the reference list), Zaccone et al. (2010) not cited in the text; We modified the reference list accordingly.

line 555, remove for these authors; This is done

line 560, These authors suggested; This is done

line 604 (Limit of detection, not acronym); This is done

line 661 LLBN? please report in full; This is done

rewrite sentence on line 675; the direct influence on the determination of Km and Vm of the use of an appropriate set...; The sentence was modified as:
'Our results clearly showed the influence of the concentration set used to compute *in situ* hydrolysis rates'

lines 679,680, 682, 683, 694 instead of affinity system I suggest affinity enzyme; This is done

line 691 Lemée 2002 or 2012?; It is 2002

line 704, supported by peptides and polysaccharides hydrolysed by enzyme activities; This is done

line 705, please report BGE in full (no acronym has been reported before) This is done

---

## Author Comment (AC2) · 20 Nov 2020

**Response to referee 2 "Spatial patterns of biphasic ectoenzymatic kinetics related to biogeochemical properties in the Mediterranean Sea" by France Van Wambeke et al. ms BG-2020-253**

**Anonymous Referee #2** **and our responses in blue**

The paper "Spatial patterns of biphasic ectoenzymatic kinetics related to biogeochemical properties in the Mediterranean Sea." by France Van Wambeke et al., reports prokaryotic ectoenzymatic activity, abundance and heterotrophic production in the epipelagic and the upper part of the mesopelagic layers in the Mediterranean Sea. In this study, the Vm and Km of the 3 enzymes (alkaline phosphatase (AP), aminopeptidase (LAP) and β-glucosidase (βGLU)) were determined using 2 series of substrate concentration. The paper points out that the choice of substrate concentrations affects the results and their interpretation.

Strength points
The paper presents an impressive quantity of data on ectoenzymatic activity of 3 enzymes: alkaline phosphatase (AP), aminopeptidase (LAP) and β-glucosidase (βGLU).
The data are of good quality and I think they deserve publication. The methods are detailed and well described. This paper has the potential to be a reference work for future studies on enzymatic activity, since the authors showed that the use of different ranges of concentrations of substrates give different results and therefore lead to bias in the interpretation of the results. I think this is a very important point, since the measure of enzymatic activity is crucial to get insights into the main biogeochemical fluxes in the oceans. Often the published data are difficult to compare due to the different concentrations of substrate used leading to contrasting interpretations.

Weak points
The main shortcoming is the language, often due to language issues, I did not understand what the authors means and the paper is confusing in many points. There are also some grammatical mistakes and typos. I think that there are too many details in the results making them hard to follow. The discussion is not focused, there are too much information that confound the reader. My main recommendation is to deeply revise the English, to simplify the results and to rework the discussion, in order to make the paper easier to follow and to highlight its main message.
We thank the referee for the suggestions and helpful comments. He/she read deeply the manuscript, checking numbers cited, tables, figure legends and provided many recommendations for style corrections and reduction of the results section as well as reorganization of the discussion, that we used to improve the manuscript and redefine various aspects of the focus of the paper as suggested. In particular:
- we shortened the description of the water masses, cited references and modified Fig.2 and Fig.8
- we simplified the results section (removed citation of numbers, DVFs too much detailed, descriptions of Figures..)
- we re-organized the discussion focusing on the interpretations of biphasic kinetics

Specific comments are reported below.

Introduction

All the introduction would benefit of a deeply revision of the English. The text does not flow well and there are some grammatical issues making hard to read it
The revised ms have been corrected by a native English speaker.

L38-40, P1. "Most of the organic matter being in the state of high molecular weight material, its hydrolysis by ectoenzymes plays an important role in the degradation, utilization and mineralization processes in aquatic environments, but also in nutrients regeneration (Hoppe, 1983; Chróst, 1991)." I think there is a grammatical issue in this sentence.
The sentence was modified as:
'In aquatic environments the organic matter compounds available for bacterial utilization are dominated by high molecular-weight organic molecules. In order to be assimilated, first they need to be hydrolyzed into smaller sized molecules by ectoenzymes outside of the cell. This represents a limiting step in organic matter degradation, and in nutrient regeneration (Hoppe, 1983; Chróst, 1991).'

L47 Seldomly?
The sentence was modified as: 'Kinetic experiments are time-consuming and most studies reporting ectoenzymatic activity examine enzyme kinetic patterns using only one or two samples. A single concentration of a presumed saturating substrate is then used to determine the activity of all the samples.'

L49-51, P2. "Within these 5 studies for the concentration kinetic the minimum concentration used was 50 nM at the lowest, ..." I think there is a grammatical issue in this sentence.
The sentence was modified as:
'Only 5 studies used a range of substrate concentrations to determine the enzyme kinetics, of these the lowest concentration used was 50 nM, (the lower concentration in the set is typically between 1 and 5 µM), and the highest concentration used was 1200 µM (the range of the higher concentration in the set is 5 - 1200 µM, median 200 µM).'

L66, P2. "among 44 isolated strains", strains of what? Bacteria?
Yes, this information was added

L95-97, P3. "The interaction between different enzymes has been largely studied in the Mediterranean Sea (Zaccone and Caruso, 2019) due to the particular role of this elemental stoichiometry." I do not understand this sentence. What do you mean with "interaction among enzymes"?
The sentence was modified as:
'Due to the particular role of this elemental stoichiometry,
the relative activities of different enzymes have been widely studied in the Mediterranean Sea (Zaccone and Caruso, 2019)'
The relative activities for instance the LAP/AP ratio, are often used as indexes of P deficiency and this is written as an example in the text of the ms.

How can it affect the C/N/P ratios of nutrients and organic matter?
If one enzyme releasing P or N shows higher activities than other types of ectoenzymes releasing only C, it is expected to see changes in the C/N/P ratios of the degraded organic matter. This will have also consequence on the fraction of N or P remineralized, as this fraction depends on the bacterial growth efficiency as well as on the relative differences in the C/N/P of organic matter relative to the C/N/P ratio of bacterial biomass.

L107-109, P3. "Our aim was to study the effects of the respective activities of the ectoenzymes in relation to the quality of the organic matter present, below the productive layer and above the deep Mediterranean waters" How was the quality of the organic matter investigated? I did not find anything about it in the paper.

We used C/N/P ratios of particulate and dissolved organic matter, TAA/DON and TCHO/DOC ratios

L104-107, P3. "In this study, we investigated in the Mediterranean Sea, the kinetics of three series of enzymes targeting proteins, phospho-mono esters and carbohydrates (aminopeptidase, alkaline phosphatase 105 and _-D -glucosidase respectively) in relation to the elemental stoichiometry of particulate and dissolved organic matter." There is only 1 line (L333, P8) about stoichiometry data in the results and few lines (L572-573 and 592-593) in the discussion, so I think that this cannot be one of the main goals of the paper.

We agree. We have removed the last part of this sentence.

L106-107, P3. "We have paid particular attention to the use of a wide range of substrates concentrations to evaluate potential multiphasic kinetics." and L116-117, P 3
"Finally, we discuss the biases in interpretation of past and current enzymatic kinetic, potentially induced by the reduced range of used substrates concentration." I think that the use of different concentration ranges of substrate for the 3 enzymes leading to different results is the main message and I would focus the manuscript on it.

We agree and we have modified this part of the introduction accordingly.
The discussion has also been reorganized as suggested by the referee. In the revised version, the end of section 4.1 now introduces the new discussion plan as follows:
'We have shown that the differences between the Km and Vm of the low and high affinity enzymes might change with the nature of the enzyme, with depth, and regionally. We will develop below the different interpretation that might emerge when discussing about i) the increase/decrease with depth ii) the use of enzymatic ratio as indicators of nutrient availability or DOM quality and iii) the estimates of *in situ* hydrolysis rates and their contribution to heterotrophic bacterial carbon or nitrogen demand.'

Accordingly, the titles of the discussion sections were modified as follows:
-       4.2 How the concentration set used affects ectoenzymatic kinetics trends with depth: possible links with access to particles,
-       4.3 How the concentration set used affects the interpretation of enzymatic properties as indicators of nutrient imbalance of DOM quality and stoichiometry.
-       4.4 How the concentration set used affects potential contribution of macromolecules hydrolysis to bacterial production

2. Materials and Methods
L143, P4. "Heterotrophic prokaryotic production (BP), heterotrophic prokaryotic abundances (BA)" I would use HPA and HPB as abbreviations.

I assume you want to write instead HPA and HPP. I recognize this would be clearer, but let me be an old generation scientist referring to the classical abbreviations BP and BA. This is just a matter of definition and both abbreviations have been properly defined in the text.
The sentence in section 2.1 was written as:
'The water sampled with the conventional CTD-rosette was used for measuring heterotrophic bacterial production (BP, *sensus stricto* referring to heterotrophic prokaryotic production), heterotrophic bacterial abundances (BA, *sensus stricto* referring to heterotrophic prokaryotic abundances)…'

L147, P4. Replace ectoenzymatic activities by EEA, since the abbreviation is defined at L143-144. This is done as well as in other parts of the manuscript

L153, 154, 156 and 157, P4. I would use upper case letters for the abbreviations, in particular for DCM, LIW and MDW.
We prefer to keep lower case letters for the 4 layers and upper case letters for biological parameters.

L156-157, P4. "and second sampled at 1000 m (the limit between meso and bathypelagic waters), except at 2 stations (FAST, 2500 m; ION, 3000 m) named 'mdw' "
Why did you select 1000 m as depth representative of deep waters and not a sample collected close to the bottom?
The first idea was to focus on the main aim of the PEACETIME cruise devoted to the impact of dust on primary production and associated fluxes. So, we focused mainly on surface and mesopelagic zones except at the 2 stations FAST and ION where it was possible to study deeper layers.

How do you think that the different sampling depth at station FAST and ION can affect the results? This should be discussed in section 4.2 and 4.3.
A sentence was added in sub-section 4.2 on this point:
'Note that for the deepest layers sampled (FAST: 2500 m and ION: 3000 m), results are also contrasting, specific AP decreases with depth at ION but increases at FAST.'

L192-193, P5. Please add the batch of the CRM you used for DOC analysis, its nominal and measured concentration.
We added the following sentence:
'The nominal and measured DOC concentrations of the two batches used in this study were 42-45 µM and 43-45 µM, for batch14-2014#07-14, and 42-45 µM and 42-49 µM, for batch17-2017 #04-17.'

L217, P5. "Bacterial production (BP, sensus stricto referring to prokaryotic heterotrophic production)", BP was already defined at L143, P4.
In the revised version, there are not anymore abbreviations neither in the abstract nor in the introduction and they are defined here in Material and Methods sub-section 2.1.

L225, P6. "On 9 occasions", do you mean replicates? Samples?
Samples. The sentence was modified as:
'On 9 occasions during the cruise transect….'

Results
The results are very heavy to read, in section 3.3 there are too many comparisons, too many details that are not relevant for the main message of the paper. I recommend to simplify this section and to avoid details not relevant for the main message of the paper.
We simplified the results section following the recommendation of the referee.

3.1 Hydrological situations.
The title should be changed, what does "Hydrological situations" mean? I suggest Physical properties

The sub-section 3.1 has been shortened following the general comment of the referee 1 with a focus on the physical properties of water masses in which the samples of enzymatic activity were collected.
The Figure 2 has been redrawn following the comment of the referee.

This section should cite some papers reporting the circulation of the Med Sea and the main physical properties of the water masses.
A short description and references on the thermohaline circulation have been added in the introduction. A reference on the identification of water masses from their physical properties has been added in this sub-section 3.1. (see response to referee 1)

L294-295, P7. "The sampled stations have basins and latitude characteristics that were superimposed on a changing the seasonal pattern", I do not understand this sentence.
This sentence has been removed in the revised manuscript.

L299, P7. "Modified Atlantic Waters (MAW) are characterized by low salinity below the seasonal thermocline" Do you mean above the seasonal thermocline?
The cruise was carried out during May-June 2017. At the period of the year (end of spring), the surface layer has been sufficiently warmed up by atmospheric fluxes to generate a seasonal thermocline. This interface separates the surface waters with the core waters of Atlantic origin: thermohaline properties remain similar in salinity, but warmer due to spring heating. MAW are located below the seasonal thermocline. This paragraph has been reformulated in this way.

L299-301, P7. "this property is stretched in the westernmost stations, then progressively relaxes on eastern station, revealing an eastward circulation in the Algerian Basin and a dispersion in the connected basins (northwestern Mediterranean, Tyrrhenian, and Ionian Seas)." And L303-305, P7. "This property is pronounced in the eastern stations and progressively lowered on the western stations, revealing an opposite circulation pattern to the MAW. " It is not possible to infer the water mass circulation from the T/S graphs. Please add references and rework the sentences.
We thank the referee for pointing out that the analysis of T/S diagrams is insufficient to infer circulation of water masses. This section has been simplified and a general description supported by references has been done in the Introduction.

L305-308, P7. Please add references.
This sub-section 3.1 was rewritten and references added as indicated above in response to referee 1.

L308-310, P8. Please add references.
This sub-section 3.1 was rewritten and references added as indicated above in response to referee 1.

L310-311, P8. "The core of LIW is characterized by lower oxygen content than its surrounding water masses, shallower (MAW) and deeper (WMDW and EMDW)". Looking at figure 2a, this observation is not always true, for example at stations ION and 6, LIW has higher oxygen than deep waters.
The referee is right. This correspondence stands for the Western Mediterranean Basin. In the Eastern Mediterranean, the oxygen minimum in LIW is local 'oxystad', its concentration

becomes larger than deep waters oxygen while getting closer to the region of LIW formation. Note that only the Figure 2b (T/S diagrams) has been kept and redrawn in the revised version

L312-314 "We thus presented all the figures/tables in the order ST10, FAST, ST1, ST2, ST3, ST4, ST5, TYR, ST6 and ION, according to the expected circulation of the LIW (from the right to the left)." If you want to follow the LIW path, I think it is better to invert the order of the stations, since LIW move from ION to St.10.
We agree with the referee that this classification is misleading with respect to LIW circulation. We kept the longitudinal classification most commonly used in the literature and applicable also for the other sampled layers/water masses. We reformulated the sentence to specify the inverted way of LIW circulation and removed the link of causality between the choice of classification and LIW circulation.

3.2 Biogeochemical situation. I would replace situation with properties. This is done
In this section, the values of DOC, DON and DOP should be reported not only their change with depth.
The ranges of DOC, DON and DOP were cited in this paragraph as well as reference to Table S1.

Section 3.3. Ectoenzymatic activities – kinetic trends
This section is really heavy and in some parts there is not correspondence between the number in the text and in the tables. The authors should carefully rework this section deleting all the details that are not relevant and checking the correspondence between the number in the text and in the tables.
This sub-section was reworked and reduced. See response to comment 'L368-L374, P9' below for the correspondence between numbers in the text and in the Tables.

L352, P8. "The ectoenzymatic activities were determined using large trophic conditions", I do not understand this sentence.
The sentence was modified as: 'EEA were determined over highly variable trophic conditions and using a wide range of substrate concentrations ranging from 0.025 to 50 µM'

L368-L374, P9. Is this paragraph relevant? Looking at table 2, I find different numbers.
We agree with your remark and reduced the paragraph accordingly.
About the 'numbers', this is a mode of computation issue. We calculated 'mean of ratios' instead of 'ratios of means'. For instance, in Table 2, I agree that Vm1APmean / Vm1LAPmean is 5.3, not 6 as cited in this paragraph. Indeed we estimated more adequate to calculate individual Vm1AP/Vm1LAP at each station, and then to compute mean of the ratios. In this example, the 10 stations had a distribution of ratios as 1.61, 3.33, 2.33, 3.95, 1.16, 1.68, 3.67, 10.56, 7.36, and 25.49 which lead to an average mean of ratios of 6.1. Note that all 'ratios' cited in the text (Vm50/Vm1, DVF, Km50/Km1, Km/Vm…. are computed as means of ratios and not as ratio of means. This sentence was added in Material and Method section.

In addition, there was a typo error in the second part of this paragraph, as we presented LAP/βGLU ratios, not AP/βGLU ratios.
To finish with this referee comment, we agree that it is necessary to reduce all these descriptions and we have reduced this paragraph to:
'For each enzyme, the order of magnitude reached for Vmax was the same at the 'surf' and 'dcm' layers. In all layers, the highest mean Vm of the 10 stations were obtained for AP,

followed by LAP and then βGLU, whatever the range of tested concentrations (Vm50 or Vm1, Table 2).'

L376-380, P9. "For LAP (Fig. 4), Vm50 was on average 3 times higher than Vm1 in 'surf' and 'dcm' layers, but the differences between these two rates increased with depth (x9 in 'liw', x12 in 'mdw'). Vm50 decreased from epipelagic to mesopelagic waters by a factor of 8 on average, (ratio 'depth variation factor' – DVF), but by a factor x19 for Vm1 (Fig. 4a)." It is very hard to see these differences in Figure 4. I think you should also refer to Table 2.
Yes, we could refer to Table 2 but as we calculated means of ratios, they cannot be directly calculated from Table 2.

Looking at table 2 the number are different. As an example:
Vm50 was 10-times not 12 higher that Vm1 in mdw. "Vm50 decreased from epipelagic to mesopelagic waters by a factor of 8 on average,", if I consider dcm epipelagic and LIW mesopelagic Vm50 decreases by 4.6 times, if I consider surf as epipelagic the decrease is of 3.8, so I don't understand how the authors calculated a value of 8, the same for Vm1.
We computed means of ratios, not ratios of means.

L383, P9. I think there is a typo SD10, SD2 and SD6 should be St10, St.2 and St.6. St. FAST and St.1 are missing, they also show Km50 of LAP lower at the dcm than in the surf.
Yes, we modified the sentence in the other way:
'Only 3 stations showed an increase at 'dcm' compared to 'surf' layers (ST3, ST4, ST5)'

L395-396, P9. Also here there is not correspondence among the numbers in the text and the numbers that I can calculate from the table. Please check.
We computed means of ratios, not ratios of means.

L410-411, P10. "Average Km50/Km1 ratio for _GLU was 320 and average Km1/Km50 ratio for LAP was 118." My calculation looking at table 4 indicate 240 instead of 320 and 79 instead of 118. Please check.
We computed means of ratios, not ratios of means.

L436, P10. I think you should cite Fig.7c,d not Fig.7a,b.
Yes, we modified the text accordingly

Discussion
The discussion is confounding, there are a lot of interesting ideas, but they are lost in the text. The discussion would strongly benefit of deeply revision of the English. I think that section 4.1 should be the focus of the paper, but I miss a conclusion. From your data do you suggest to use Vm1 or Vm50? Km1 or Km50? Or since they give different information does the use of one or the other depend on the goal of the work? From my understanding the use of a not-appropriate range of substrate determine bias in the results and in the interpretation and data, obtained using different range of substrate, are not comparable. I think these points should be better stressed in this section. I also think that the other sections should support this one, showing how the different ranges may affect the interpretation of trends with depth and regional variability.
We agree with referee 2 comments. This has been also suggested by reviewer 1, see our response above to 'L106-107, P3'

L481, P11. "The biphastic factor as defined in Tholosan et al (1999). "Please define it in the text to help the reader.

We changed the term of biphasic factor to biphasic indicator in order to use the same as in Tholosan et al. (1999).

The definition of the term was introduced in sub section 3.3 when we describe turnovertimes as:

'We estimated the degree of difference between the kinetics based on low or high concentrations sets using the "biphasic indicator" as developed in Tholosan et al. (1999). This index tracks the difference between the initial slopes (Vm/Km) of Michaelis-Menten kinetics as (Vm1/Km1) / (Vm50/Km50).'

Then the term was also cited in the discussion sub section 4.1 where the sentence was modified as:

'In our study the biphastic indicator (Km50/Vm50) / (Km1/Vm1) was used to determine the degree of difference between the two Michaelis-Menten LAP kinetics'.

L505-509, P12. Please rework, I do not understand this sentence.

The sentence was modified as:

'Conversely to AP results, the higher differences between the 2 LAP enzymatic systems, suggest that microorganisms expressing LAP activity faced large gradients of protein concentrations and were adapted to pulsed inputs of particles.'

L535-539, P13. This paragraph should be moved in section 4.1.

The discussion section was re-organized as suggested by both referees

L569-571, P13. "With concentration kinetics ending at 50 μM of MUF-P, the specific activities of AP reached using per cell Vm50 or per cell Vm1 were not so different and their trend with depth were similar (Fig. 8). " It is really hard to see these trends in Fig.8.

We modified the sentence as:

'We could not conclude that there was a systematic increase of specific AP with depth. Specific AP decreased in 5 stations, increased in 3 stations and at the remaining stations specific Vm increased based on Vm50, but decreased based on Vm1 (Fig. 8b).

L572-573, P14. "whereas DOC/DOP ratio decreased (from 2200-2400 to 1500-1200), suggesting a preference for heterotrophic prokaryotes to use dissolved organic phosphorus as substrate of AP." Usually DOC/DOP ratio increases with depth due to the preferential removal of P. Are these ratios calculated using data collected in this cruise? The removal of DOP by heterotrophic prokaryotes should increase the DOC/DOP so, this sentence has no sense to me.

We agree with the referee comment about an expected DOC/DOP increase with depth. Yes we measured DOP and DIP during this cruise however we have no DOP data at ST1, 2 and 10. In addition, DOP concentration at depth is on average 39 nM, and as it is obtained by a difference between TDP and DIP with average values around 350 - 400 nM, its estimate is subject to large errors. Thus we had only few data on DOC/DOP ratios at depth (6 in liw and 2 in mdw) associated with large errors. Instead of describing means per depth, which is subject to caution under such conditions, we examined in more detail variation with depth at all the stations. We found an increase of DOC/DOP ratio in liw or mdw layers in 2 cases (ST2 and ST4), a decrease in 4 cases (ST3, ST5, ST6 and ION), and the same order in one case

(TYR). Note that this observation is obtained from only 4 layers sampled along the profile, and such trend would be much easier based on a complete profile rather on only 4 data points.

We added a sentence in the results section, sub section 3.2 as:
'Taking all 4 water layers, the mean values for the DOC/DON and DOC/DOP molar ratios were 14 ± 2 and 2112 ± 1644, respectively, with no significant trend with depth due to the variability within stations. Deep DOP was not sampled for at 3 stations and in addition DOP estimates are subject to large errors at depth (DIP is on average 10 times higher than DOP). We observed a DOC/DOP increase with depth at 4 stations, but a decrease at 2, and no trend at another.'

We modified the sub sub section 4.2 the discussion on this topic as:
'The particulate matter C/P ratio did not change with depth. However the variability in the trend with depth seen for specific AP was also observed with DOC/DOP ratio. We expected to see an increase with depth due to a preferential removal of P, however, it was not systematic.'

L574-587, P14. This paragraph should be moved in section 4.1.
The discussion section was re-organized as suggested by both reviewers

L595-611, P14. This paragraph is not clear to me.
This paragraph was reorganized and we hope it is clearer now

4.3 Regional variability
L613-615, P14. "In epipelagic waters, both AP maximum rates (Vm1, Vm50) significantly increased from the Algerian/Ligurian Basins to the Tyrrhenian Basin (t test, p = 0.002 and p = 0.02, respectively), and reached maximum values at ION.". In Figure 6 this pattern is not very clear.
Yes this is because the Fig 6 is in log scale, more adapted to compare AP rates between epipelagic and deep layers. The factor of increase between the 2 regions was about 3 (x3.3 for Vm50, x 2.6 for Vm1)
The sentence was modified as:
'In epipelagic waters, both AP maximum rates (Vm1, Vm50) significantly increased by approx 3 fold from the Algerian/Ligurian basins to the Tyrrhenian basin….'

L658-663, P15-16. This paragraph should be moved in section 4.1.
The discussion section was re-organized as suggested by both referees

L675-687, P16. This paragraph should be moved in section 4.1.
The discussion section was re-organized as suggested by both referees

Figure 2. Please enlarge the name of water masses. I think the value of this figure would strongly increase if you mark on the T/S graphs where the samples for enzymatic activity were collected. You could also add a zoom of intermediate and deep waters.
We followed the Referee's suggestion. The Figure 2 has been redrawn in the revised version with a zoom on the densest waters and the addition of samples marks as follows:

[Figure]

Figure 4-6. Please add the legend.
This is done

Figure 8. This figure is really confuse and hard to understand, I suggest to remove it.
We separated the different enzymes in separate plots as follows:

[Figure]

Figure 9 and 10. I suggest to report in the caption how bacterial nitrogen demand and bacterial carbon demand was calculated.
For Fig. 9 we modified the legend as:
 '….and comparison to heterotrophic bacterial nitrogen demand, determined from BP assuming a biomass C/N molar ratio of 5 and no active excretion of nitrogen….'

For Fig. 10 we modified the legend as:
 '…heterotrophic bacterial carbon demand (BCD, determined from BP assuming a BGE of 10%) in epipelagic ….'

Table1, Please check the title of the columns.
This has been done. Two blank missing (dcm depth, bottom depth) and the term 'layers' for the two last columns.

Table 2, line 6 column 5, I think the lines are inverted, if not please check the number in the text.
Yes mean sd and min max were inverted, we corrected.

---

## Author Response (AR1)

**Response to Christina Klass Co-Editor-in-Chief ms BG-2020-253**

Dear Dr Klass

First, we acknowledge that your comments are fully justified and have encouraged us to modify the data presentation accordingly. You will find below a step by step response to your comments/recommendations.

*the written text requires quite some editing*

You made a big effort to improve from page 1 to 9 a ms which was only the first version of the ms. We apologize for this. Indeed, the BGD process did not allow to submit the corrected version in the first round of review (which had been edited in English as well), only the answers to the 2 reviewers have been posted. We have now improved this revised version with your suggestion, please find the attached document. All typos have been corrected, we underlined in yellow the main modifications made in the ms.

*nowhere in the data and analysis it presents robust evidence of biphasic kinetics.*

You are right, this was not sufficiently detailed. We added new paragraphs in the methods and in the results section and 2 additional figures (Figure 4, Figure S2) and one table (Table S2). For assessing the presence of biphasic kinetics statistically, we used the F test presented in Tholosan et al. (1999). The lines 287 - 303 in the method section describe the statistical tool, and the results of statistics are now presented on lines 362 - 411 and on table S2. We added also a new paragraph at the beginning of the discussion (lines 518 - 526).

*This is in contrast to previously cited work such as Tholosan et al. (1999; with the use of Lineweaver-Burk plots)*

Note that Tholosan et al. (1999) used the Lineweaver-Burk plots to illustrate evidence of biphasic kinetic but they used non linear regression fits from the Michaelis-Menten kinetic to make their statistics. We plotted you below the Lineweaver-Burk plots fitted to the data set presented in Figure 3 (full lines Lineweaver-Burk fit for the 2.5-50 µM concentration set; dotted lines fit for the 0.025-1 µM concentration set). We found that this representation, in the frame of our concentrations ranges (the inverse of a 0.025-1 µM concentration set to be compared to the inverse of a 2.5-50 µM set) is not easy to visualize, and therefore choose not to present such plots in the revised ms.

[Figure]

*the standard errors presented (which in the case of your manuscript are also too large to corroborate the existence of two enzymatic systems).*

The % variation of the standard errors of the Km and Vm values are listed on Table S2 and standard errors are visible on Figure 4 in which plots are not in log scale. With all these new information and tables, we prove the existence of a biphasic system in 60% of the cases.

*Further, if enzymes operate at different range of concentrations, it is contradictory to use the whole data range for the estimates of the low affinity system.*

In the first part of the results section, we present now the 3 series of data set. Note that the $Km_{50}$ and $Vm_{50}$ terms in this revised version corresponds to kinetic parameters derived for a 2.5-50 µM concentration range whereas the terms $Km_{all}$ and $Vm_{all}$ are now describing the kinetic parameters derived from a 0.025-50 µM, i.e whole concentration range. We demonstrate that, linked to the uncertainty in the distribution of the data points used to describe Michaelis-Menten kinetics, the kinetic parameters obtained by using the global model or the model 50 were not so different, and at least still very different than those obtained by using the model 1. We also showed on Figure S2 that a series of Michaelis-Menten kinetics can be obtained by addition of successive increasing concentration in the data

set. Finally, we decided not to focus the discussion *sensus stricto* on biphasic kinetics, but rather on the consequences of using a concentration set restricted to low concentration (up to 1 µM) in comparison to most published studies that use a concentration set reaching much higher concentrations (see the first part of the discussion), and for this we considered that the global model was more representative. The title of the ms was also modified accordingly.

---

## Editor Decision (ED1)

**Spatial patterns of biphasic ectoenzymatic kinetics related to biogeochemical properties in the Mediterranean Sea.**

France Van Wambeke1, Elvira Pulido1, Julie Dinasquet2,3, Kahina Djaoudi1,4, Anja Engel5, Marc Garel1, Sophie Guasco1, Sandra Nunige1, Vincent Taillandier6, Birthe Zäncker6,7, Christian Tamburini1.

[revised manuscript text omitted]
 $\pm$ sd               | $1.00\pm0.78$   | $1.20\pm0.92$   | $0.26\pm0.24$     | $0.17\pm0.13$   |
| nmol l -1 h -1 | range                       | 0.36 - 2.85     | 0.33 - 2.83     | 0.08 - 0.91       | 0.06 - 0.45     |
| Vm1 LAP                              | mean $\pm$ sd               | $0.29\pm0.10$   | $0.45\pm0.25$   | $0.028 \pm 0.014$ | $0.017\pm0.010$ |
| nmol $l^{-1} h^{-1}$                 | range                       | 0.21 - 0.56     | 0.19 - 0.98     | 0.014 - 0.060     | 0.007 - 0.042   |
| Vm50 βGLU                            | $mean \pm sd$               | $0.13\pm0.04$   | $0.12\pm0.08$   | ld                | ld              |
| nmol $l^{-1} h^{-1}$                 | range                       | 0.08 - 0.23     | 0.03 - 0.30     |                   |                 |
| Vm1 βGLU                             | $mean \pm sd$               | $0.019\pm0.009$ | $0.025\pm0.019$ | ld                | ld              |
| nmol $l^{-1} h^{-1}$                 | range                       | 0.012 - 0.040   | 0.014 - 0.077   |                   |                 |
| Vm50 AP                              | $mean \pm sd$               | $2.56\pm2.58$   | $3.73 \pm 4.52$ | $0.38\pm0.48$     | $0.25 \pm 0.40$ |
| nmol $l^{-1} h^{-1}$                 | range                       | 0.30 - 8.30     | 0.11-14.6       | 0.04 - 1.66       | 0.06 - 1.30     |
| Vm1 AP                               | $mean \pm sd$               | $1.55 \pm 1.58$ | $3.01 \pm 4.01$ | $0.02 \pm 1.11$   | $0.12\pm0.25$   |
| nmol $l^{-1} h^{-1}$                 | range                       | 0.25-5.62       | 0.07-13.2       | 0.24 - 0.33       | 0.01 - 0.80     |
| Km50 LAP                             | mean $\pm$ sd               | $7.4\pm 6.9$    | $5.2\pm7.7$     | $17.8 \pm 15.2$   | $22.3\pm26.3$   |
| μΜ                                   | range                       | 0.8–20.9        | 0.4–25.0        | 3.6-41.6          | 1.8 - 83.8      |
| Km1 LAP                              | $mean \pm sd$               | $0.50\pm0.20$   | $0.42\pm0.28$   | $0.23 \pm 0.19$   | $0.25{\pm}0.27$ |
| μΜ                                   | range                       | 0.12-0.83       | 0.07–0.90       | 0.10 - 0.69       | 0.01 - 0.88     |
| Km50 βGLU                            | $\text{mean} \pm \text{sd}$ | $10.6\pm6.3$    | $8.7\pm7.2$     | ld                | ld              |
| μΜ                                   | range                       | 4.4–27.4        | 1.2–24.3        |                   |                 |
| Km1 βGLU                             | $mean \pm sd$               | $0.044\pm0.071$ | 0.11±0.11       | ld                | ld              |
| μΜ                                   | range                       | 0.009-0.244     | 0.01 – 0.36     |                   |                 |
| Km50 AP                              | $mean \pm sd$               | $0.72\pm0.71$   | $0.49\pm0.34$   | $2.25\pm2.42$     | $3.7 \pm 6.8$   |
| μΜ                                   | range                       | 0.09–2.18       | 0.18-1.07       | 0.17 - 7.32       | 0.4 - 21.9      |
| Km1 AP                               | $mean \pm sd$               | $0.11\pm0.03$   | $0.27\pm0.28$   | $0.37\pm0.22$     | $0.27\pm0.16$   |
| μΜ                                   | range                       | 0.07–0.14       | 0.05 - 0.80     | 0.14 - 0.89       | 0.06 - 0.52     |
| BA                                   | $mean \pm sd$               | $5.3\pm1.6$     | $5.4 \pm 1.5$   | $1.13\pm0.40$     | $0.56\pm0.15$   |
| $10^5$ cells ml -1        | range                       | 2.1-7.8         | 4.0 - 8.5       | 0.41 - 1.91       | 0.33 - 0.78     |
| BP                                   | mean $\pm$ sd               | 37 ± 13         | $21 \pm 7$      | $0.77 \pm 0.40$   | $0.27 \pm 0.19$ |
| ng C $l^{-1} h^{-1}$                 | range                       | 26 - 64         | 12 - 32         | 0.39 - 1.60       | 0.07 - 0.60     |

| anzyma unite durfaça dem liur mdur DVE | deep waters (low and mdw data). The distribution of cell specific Vm1 and cell specific Vm50 for AP and LAP are also pres | DVF is the 'depth decreasing factor', calculated for each station as mean value in epipelagic water (surface and dcm data) div | production (per unit BP)and iii) particulate organic matter: nitrogen (PON) for LAP, carbon (POC) for βGLU and phosphoru | fluorogenic substrates; Vm1), and specific to either i) abundance of total heterotrophic prokaryotes (per cell activities), ii) he | Table 3 Range of different potential specific activities calculated using vm derived from the low range of concentration teste |
|----------------------------------------|---------------------------------------------------------------------------------------------------------------------------|--------------------------------------------------------------------------------------------------------------------------------|--------------------------------------------------------------------------------------------------------------------------|------------------------------------------------------------------------------------------------------------------------------------|--------------------------------------------------------------------------------------------------------------------------------|
|                                        | LAP are also presented on Fig 7.                                                                                          | and dcm data) divided by the mean in                                                                                           | U and phosphorus (POP) for AP.                                                                                           | l activities), ii) heterotrophic bacterial                                                                                         | concentration tested (25 to 1000 nM                                                                                            |

| enzyme             | units                                                            | surface     | dcm         | liw       | mdw       | DVF       |
|--------------------|------------------------------------------------------------------|-------------|-------------|-----------|-----------|-----------|
| Per cell LAP       | 10 -18 mol leu bact -1 h -1     | 0.33-1.52   | 0.44-2.18   | 0.11-0.70 | 0.12-0.54 | 1.3-9.6   |
| Per cellβGLU       | 10 -18 mol glucose bact -1 h -1 | 0.02 - 0.11 | 0.02 - 0.17 | nd        | nd        | nd        |
| Per cell AP        | 10 -18 mole P bact -1 h -1      | 0.45-26     | 0.11-32     | 0.13-11   | 0.17-23   | 0.1 - 28  |
| Per cell BP        | 10 -18 g C bact -1 h -1         | 46-136      | 25-60       | 3-17      | 1-14      | 4-23      |
| per BP LAP         | nmol AA nmol C -1                                     | 0.04-0.24   | 0.12-0.44   | 0.21-1.08 | 0.36-3.03 | 0.09-0.76 |
| per BP $\beta$ GLU | nmol glucose nmol C -1                                | 0.003-0.017 | 0.007-0.034 | nd        | nd        | nd        |
| per BP AP          | nmol P nmol C -1                                      | 0.09-2.3    | 0.05-11     | 0.46-8    | 0.6-40    | 0.04-1.7  |
|                    |                                                                  |             |             |           |           |           |

| Table 4. Turnover times-of ectoenzymes (Km/Vm ratio). Means ± sd and range values given       |
|-----------------------------------------------------------------------------------------------|
| for all stations (n = 10). For leucine aminopeptidase (LAP), beta glucosidase ( $\beta$ GLU), |
| alkaline phosphatase (AP). Id limits of detection, not enough data to plot Michaelis Menten   |
| kinetics. The turnover times are calculated from concentration kinetics using up to 50 µM     |
| concentration or up to 1 µM concentration.                                                    |

|                | Units: days                 | surface         | dcm             | liw layers      | mdw waters      |
|----------------|-----------------------------|-----------------|-----------------|-----------------|-----------------|
| Km50/Vm50 LAP  | $\text{mean}\pm\text{sd}$   | $309 \pm 214$   | $144 \pm 186$   | $2912 \pm 1756$ | $4526\pm2118$   |
|                | range                       | 94-880          | 40-663          | 1294-6616       | 1308-7791       |
| Km1/Vm1 LAP    | $\text{mean} \pm \text{sd}$ | $75\pm28$       | $41 \pm 22$     | $345\pm235$     | $634\pm713$     |
|                | range                       | 15-120          | 15-82           | 141-985         | 55-2481         |
| Km50/Vm50 βGLU | $mean \pm sd$               | $3464 \pm 1576$ | $3091 \pm 1551$ | ld              | ld              |
|                | range                       | 1997-7395       | 328-5481        | nd              | nd              |
| Km1/Vm1βGLU    | $mean \pm sd$               | $126\pm233$     | $247\pm273$     | ld              | ld              |
|                | range                       | 20-784          | 15-873          | nd              | nd              |
| Km50/Vm50 AP   | $mean \pm sd$               | $18\pm20$       | $39\pm46$       | $563\pm542$     | $1042 \pm 1156$ |
|                | range                       | 2-69            | 0.7 – 113       | 16 – 1441       | 20 - 3875       |
| Km1/Vm1 AP     | mean $\pm$ sd               | $5.6 \pm 5.0$   | $27 \pm 37$     | $268 \pm 349$   | 301 ± 172       |
|                | range                       | 1-17            | 0.6-106         | 12-1180         | 14-594          |

---

## Author Response (AR2)

**Response to Christine Klass "Spatial patterns of biphasic ectoenzymatic kinetics related to biogeochemical properties in the Mediterranean Sea" by France Van Wambeke et al. ms BG-2020-253**

Dear author,
The revised manuscript has tackled most of the main comments from previous versions in a convincing manner. There are still a few minor issues that need addressing. See comments below and annotated manuscript. The annotated manuscript also contains suggestion for improvement of the text which needs a thorough revision. I would urge you to take these comments into consideration (this has not always been the case in the last submitted version). I would accept publication of your manuscript pending these minor revisions.
Sincerely,
Christine Klaas

We would like to thank Dr Klass for her thoughtful comments and recommendations, which we address below in blue. We also took the recommendations from the annotated pdf into account in the revised version of our manuscript.

Comments (further comments are given in the attached annotated manuscript):

Line 381-382: referring to figure 4, which only shows one example for the argument is not sufficient.
We guess that there is a misunderstanding here, as the example is shown in Figure 3. On these lines, we refer to Figure 4, which shows the plots $Vm_{50}$ vs $Vm_{all}$ and $Vm_1$ vs $Vm_{all}$ (and the same for Km) for the 3 enzymes. We also modified this paragraph (see below).

Similarly, in the following statements (whole paragraph, lines 382-388) you discuss the individual model parameters and their se, but in Table S2 present average values for each layer over several stations instead of the values discussed. Please show the actual data discussed here. I suggest that in supplement Table S2, instead of statistical data on average parameters for each layer, which is already partially shown in Table 2, you provide the individual Km1, Vm1, Km50, Vm50 and Kmall, Vmall with their uncertainties as estimated for each station and layer.
We understand this editor concern. However, showing all values will require a huge table including 10 stations, 4 layers, 3 enzymes, 3 models and 2 kinetic constants (and their errors). We chose to make the table simpler as the whole data set is available upon request in the INSU/CNRS LEFE CYBER database, as indicated in the data availability section of the ms. Nevertheless, for more clarity, this paragraph (lines: 528-534 of the marked-up revised version) was rephrased to better fit with Table S2, it now reads:
'For LAP and βGLU, $Vm_{all}$ and $Vm_{50}$ were close, and the distribution of these data fitted to the 1:1 axis (Fig. 4). For LAP and AP, $Vm_{50}$ were subjected to higher errors than those of their corresponding $Vm_{all}$ (Fig. 4), as the percentage of standard error (se%; Table S2) of $Vm_{50}$ was higher than that of $Vm_1$ in most cases (40/40 for LAP, 24/25 for AP). At the opposite, for βGLU se% was higher only in 6 out of 20 cases. The relationships between $Km_{50}$ and $Km_{all}$ showed the same trend, although $Km_{50}$ were generally slightly higher than their corresponding $Km_{all}$, in particular for βGLU. As noted for Vm, the se% was higher for $Km_{50}$ than for $Km_1$ in most of the cases for LAP (39/40) and AP (25/25) and the opposite was seen for βGLU (5/20).'

I have the same comment for the next paragraph (lines 389-400), please show the individual values for the biphasic indicator in a figure or in the supplement Table 2.
We agree with this comment. We added 3 lines to the Table S2 (one per enzyme), to show the range of biphasic indicator for each layer.

In the text we developed the descriptions of biphasic indicators as follows (lines 598-604 of the marked-up revised version):
'The biphasic indicator was particularly marked for βGLU (means of 87 in SURF and 47 in DCM layers), but it was highly variable (Table S2). For LAP the mean indicator increased from ~9 in SURF and DCM layers to ~16 within LIW and MDW layers, however due to its high variability (Table S2) this increase was insignificant. For AP the biphasic indicator remained constant ($p > 0.05$) between the epipelagic layers (mean of 12 in SURF and of 6 in the DCM) and the deeper layers (mean of 5 in LIW and 9 in the MDW), with overall lower variability than for the 2 other enzymes, Table S2).'

p.16 includes a detailed discussion on Vm ratios between different enzymes, yet the data is presented nowhere. It would be helpful to include a figure similar to Fig. 9.
We agree with the Editor. A new figure showing AP/LAP and LAP/βGLU enzymatic ratios was added in the supplement material (please see Fig. S4).

Discussion lines 729-731 and Fig 11: It seems odd when you have BP data measured directly to include a 10% conversion efficiency factor. The BP is the actual carbon demand. The 10% factor would only make sense if you want to concert BP into growth rates or biomass accumulation. This needs to be corrected
We do not fully agree with this comment. Bacterial carbon demand (BCD) is the sum of bacterial respiration and bacterial production (BCD=BP+BR), and BCD=BP/BGE, BGE being the bacterial growth efficiency. As we don't know the fate of carbon in amino acids or carbohydrates after their uptake inside the cells (anabolism or catabolism), we need to compare the carbon sources issued from LAP and BGLU hydrolysis to BCD and not to BP. For the N demand, as we stated, if we assume no N excretion, heterotrophic bacterial N demand can be directly calculated from BP using the C/N ratio of 5.
As in the Piontek et al. (2014) study the contribution of hydrolysis was only was related to BP, therefore we used our value of BGE 10% to estimate a contribution of hydrolysis to BCD, in order to be able to compare the data of this study with our data set.

in Figures 2, 5-8 and 10-11 please use capital to indicate layers (SURF, DCM, LIW, MDW) as in the text.
Done

Figure 5b: x axis legends are missing
Corrected

Figure 9: in plot legends replace "per cell Vmall etc..." with "specific" or "cell specific Vmall " or Vm1, respectively. Or find a different abbreviation for cell specific Vm. (see annotated manuscript).
We introduced the abbreviation cs as suggested, in Fig. 9 and Fig 8.

Supplement Table S1: replace ld with "<dl" and explain the abbreviation in the legend: "<ld: below detection limit"
Done

Change legend Supplement Figure S2 to: "Nonlinear least squares regression fits of Michaelis-Menten kinetics for different and incremental ranges of substrate concentrations from 0.25 corresponding to a 0.025-0.25µM substrate concentration set to 50 corresponding to a 0.025-50µM substrate concentration set for a) LAP, b) βGLU and c) AP. Red dots correspond to the field measurements. The dataset is the same as in Figure 3 (DCM at station FAST). d, e f : corresponding distribution of the Vm and Km parameters calculated according to the different concentrations ranges tested."
Done

Supplement Figure S2: It would be helpful if the colors for the dataset used in panels a, b, c (0.25,05, 1 etc...) were used for the corresponding Vm and Km values in panels d,e,f.
The color codes of the plots presented in a, b, c correspond to the different concentrations indicated by the x axis in d, e, f. To be clearer, the last part of the legend was modified as: 'd, e f : corresponding distribution of the V and Km parameters plotted according to the maximum concentration added.'

Please find below the responses to some of your comments in the annotated version of the ms

Lines 438 : '…AP was the enzyme for which $Vm_1$ and $Vm_{all}$ were the closest (average of $Vm_{all}$ /$Vm_1$ ratio for the whole data set was 1.9 ± 1.2) (Fig. 7a), confirming that saturation rates occurred with 1 µM MUF-P addition … ' I do not understand how the values of V are an indicator for the level at which the rates saturate. If anything the argument should be based on the Km value or the slope (dV/dS) of the kinetic response
The sentence was reformulated (lines 634-637 of the marked-up revised version) as:
'AP was the enzyme for which $Vm_1$ and $Vm_{all}$ were the closest (average of $Vm_{all}$ /$Vm_1$ ratio for the whole data set was 1.9 ± 1.2) (Fig. 4c, 7a). Fits to model 50, using 2.5 to 50 µM concentration sets were often not significant (Table S2), because the rates stayed constant when adding these concentrations.'

Line 454 about turnovertime. Can you please explain were this this concept comes from (reference?). I understand the term turnover number (but to estimate it you also need to know the enzyme concentration).
The turnover time is defined by the ratio Km/Vm (conversely, Vm/Km defines turnover rate) and it has often been described and cited in different studies (e.g. Tholosan et al, 1999, Van Wambeke et al., 2002; Crottereau and Delmas, 1998, Unanue et al 1999; Misic et al., 2002). These ratios are used to estimate the ability of ectoenzymatic systems to be competitive at a low substrate concentration and were initially described in studies about monomer uptake or growth (Healy et al., 1980; Wright and Hobbie, 1966). We added a sentence in M&M section (line 362 in the marked-up revised version) as follows: 'The turnovertime was estimated as the ratio Km/Vm (Wright and Hobbie, 1966).'

Lines 712-716 about bacterial carbon demand and growth efficiency. 'Again clumsy and unnecessary'
These lines are not a repetition of results. In the results we describe in situ hydrolysis rates (section 3.5), but not nitrogen demand or bacterial carbon demand which are discussed here (paragraph starting line 1294 in the marked-up revised version).

Line 734 '…..as some cyanobacteria can also express LAP….' : what is the point here.

Part of the activity could be due to photosynthetic organisms, not only hprok, thus the TAA hydrolysis flux is probably not only devoted to heterotrophic bacterial N demand

The sentence was modified (lines 1387-1390 in the marked-up revised version) as:

'In our study, the contribution of TAA hydrolysis to bacterial N demand was higher in the DCM than in the SURF (10 to 40% based on the high affinity enzyme). Nevertheless, this calculation may be biased as not only heterotrophic microorganisms but also autotrophic cyanobacteria such as *Synechococcus* and *Prochlorococcus,* which are dominating phytoplankton groups in the Mediterranean Sea (Siokou-Frangou et al., 2010), can also express LAP (Martinez and Azam, 1993) to satisfy their N requirement.'

References not cited in the ms

Healey, F. P.: Slope of the Monod equation as an Indicator of Advantage in Nutrient Competition, Microb. Ecol., 5, 281-286, 1980.

---

## Editor Decision (ED3)

[revised manuscript text omitted]

590 removal of P,  was not

LAP enzymatic systems showed more  trends with depth,  but cell-specific LAP showed contradictory results: at all stations cell-specific Vm$_1$ decreases with depth (according to the DVF criterion, Fig. 9a) whereas Vm$_{all}$ remained stable (2 stations over 10) or increased with depth (5 stations over 10).

595 Using a high concentration of MCA-leu other authors have  found an increase in LAP activity per cell with depth in bathypelagic layers (Zaccone et al., 2012; Caruso et al., 2013).

 Baltar et al. (2009b), using a concentration of substrates ranging from 0.6 to 1200 μM,

600 reported an increase in the LAP  (~400 to 1200 μM) and AP  (~2 to 23 μM) with depths down to 4500 m in the sub-tropical Atlantic. In contrast, Tamburini et al. (2002), using a concentration of substrates ranging from 0.05 to 50 μM, obtained lower Km values (ranging between 0.4 and 1.1 μM) for LAP in the Mediterranean deep waters (down to 2000 m depth). It is however difficult to come to a conclusion about the effect of the concentration on Km variability

605 with depth by comparing 2 studies from different environments and using different sets of substrate concentrations. In our study where both kinetics were determined in the same waters, Km$_{all}$ in particular, increased with depth more than Km$_1$, and the ratio Km$_{all}$/Km$_1$ switched from ~16 in

epipelagic waters to 121 and 316 in LIW and MDW layers, respectively. From our data set, among the two parameters  Vm and  Km,  Km  showed the  differences

610 between the 2 types of kinetics. At many stations (TYR, ION, FAST and ST10),  $Km_1$ was stable or decreased with depth whereas  $Km_{all}$ increased, suggesting that within deep layers LAP activity was linked more to the availability of suspended particles or fresh organic matter  sinking material, than to DON. Thus, the difference between $Km_1$ and $Km_{all}$ might reflect  to  spatial and/ or temporal patchiness in the distribution of

615 suspended particles. Freshly sinking material , because of the small volume of water , but could contribute  free bacteria, small  particles and DOM  associated  (
[revised manuscript text omitted]

[Figure]

Fig 1

[Figure]

Fig 2

[Figure]

Fig 3

[Figure]

Fig 4

[Figure]

Fig 5

[Figure]

Fig 6

[Figure]

Fig 7

[Figure]

Fig 8

[Figure]

Fig 9

[Figure]

Fig 10

[Figure]

Fig 11